# Overcoming the design, build, test bottleneck for synthesis of nonrepetitive protein-RNA cassettes

Noa Katz[1], Eitamar Tripto[2], Naor Granik[1], Sarah Goldberg[1], Orna Atar[1], Zohar Yakhini [3,4], Yaron Orenstein[5] & Roee Amit [1,6✉]

We apply an oligo-library and machine learning-approach to characterize the sequence and structural determinants of binding of the phage coat proteins (CPs) of bacteriophages MS2 (MCP), PP7 (PCP), and Qβ (QCP) to RNA. Using the oligo library, we generate thousands of candidate binding sites for each CP, and screen for binding using a high-throughput dose-response Sort-seq assay (iSort-seq). We then apply a neural network to expand this space of binding sites, which allowed us to identify the critical structural and sequence features for binding of each CP. To verify our model and experimental findings, we design several non-repetitive binding site cassettes and validate their functionality in mammalian cells. We find that the binding of each CP to RNA is characterized by a unique space of sequence and structural determinants, thus providing a more complete description of CP-RNA interaction as compared with previous low-throughput findings. Finally, based on the binding spaces we demonstrate a computational tool for the successful design and rapid synthesis of functional non-repetitive binding-site cassettes.

[1] Department of Biotechnology and Food Engineering, Technion - Israel Institute of Technology, Haifa, Israel. [2] Department of Biomedical Engineering, Ben-Gurion University of the Negev, Beer-Sheva, Israel. [3] Department of Computer Science, Technion - Israel Institute of Technology, Haifa, Israel. [4] School of Computer Science, Interdisciplinary Center, Herzliya, Israel. [5] School of Electrical and Computer Engineering, Ben-Gurion University of the Negev, Beer-Sheva, Israel. [6] Russell Berrie Nanotechnology Institute, Technion - Israel Institute of Technology, Haifa, Israel. ✉email: roeeamit@technion.ac.il

  

For the past two decades, synthetic biologists have built a portfolio of increasingly sophisticated biological circuits that are able to perform logical functions inside living cells[1–4]. Such circuits are made from "biological parts" which are biochemical analogs of electronic components that are routinely used for the design of electrical circuits. Unfortunately, unlike their electronic counterparts, connecting biological parts to form circuits often fails. This is mostly due to the fact that many parts are short sequences of DNA or RNA, and connecting them introduces unpredictable and undesirable sequence effects[5]. As a result, many iterations of trial and error are often needed before a successful design is achieved. This is termed the design, build, test (DBT) cycle in synthetic biology and is considered to be a major bottleneck for progress in the field. Specifically, the field is lacking computational methods that allow users to reliably design their system of choice without going through multiple time-consuming DBT cycles. The challenge of formulating such algorithms is rooted in the large space of biomolecules, and the variety of possible interactions between them. This translates to a plethora of molecular mechanisms, each governed by different kinetics, thermodynamic parameters, and free-energy considerations. Consequently, modeling these systems necessitates case-specific kinetic and/or thermodynamic modeling approaches. Reliable algorithms are especially needed for the design of RNA-centric functional modules, and for transcriptional expression systems.

In recent years, the emergence of oligo-library (OL) studies[5–9] have provided the community with increasingly large databases of characterized synthetic and natural parts (e.g., enhancers, promoters and 5′ UTR segments) in various cell types. This plethora of data allows one to devise quantitative thermodynamic or machine learning (ML) models, which can then be tested on predicted sequences to assess their reliability. Several works have been published demonstrating that OLs of promoters in *S. cerevisiae* and *E. coli*[10–12] (numbering several hundred thousand to millions of variants) provide a suitable training set for ML algorithms. The performance of these algorithms on unseen data as quantified by Pearson correlation was ~0.8. In another recent study, the authors used multiple rounds of experimentation and variant analysis to devise an algorithm for the functional design of non-repetitive sgRNA cassettes in bacteria[13]. Therefore, a large dataset of characterized parts is a prerequisite for devising a reliable design algorithm.

Another RNA-based system, where a reliable design algorithm can be useful for various applications, are phage coat proteins (CPs). These RNA-binding proteins (RBPs) have been utilized as both a model system for understanding RNA–protein interactions, as well as for a variety of applications, including gene editing and RNA-tracking[14–19]. Typically, these proteins are utilized in conjunction with a synthetic long non-coding RNA cassette that encodes multiple repeats of the desired CP binding sites. However, a limited understanding of CP-binding in vivo has forced cassette designs to incorporate repeated hairpin-like sequence elements, making them cumbersome to synthesize using current oligo-based technology. Subsequent steps, including cloning and genome maintenance, are also badly affected by the repeat nature of the cassette. Moreover, repeat sequence elements are notoriously unstable[20], thus damaging protein binding to the cassette and causing occupancy-related experimental noise. Consequently, these limitations hinder the utility of these cassettes for robust quantitative measurements[21] as well as expansion to more complex multi-genic applications.

Previous findings have indicated that specificity in phage CP binding to RNA is determined by structural elements formed by specific sequence motifs[22–28]. This implies that for a given phage CP, many different sequences may become potential binding sites by folding into a common functional structure. These characteristics

make CP binding to RNA a suitable candidate for an OL experiment. However, while useful for identifying functional variants, the OL scale is much smaller than the available sequence space for ~20–25 nt-long binding sites. Thus, the vast majority of functional variants cannot be sampled. Recently developed ML algorithms[29–31] provide the necessary tool to computationally expand the variant database to millions of potentially functional sequences, using the OL as an empirical training dataset. The result is a ML model that can computationally score any sequence for the desired functionality, and thus provide a more complete description of the characteristics of CP binding to RNA.

In this work, we apply the OL–ML approach to characterize the sequence and structural determinants of phage CP RNA binding sites. We generate an OL of ~20,000 candidate sites for the phage CPs of MS2 (MCP), PP7 (PCP), and Qβ (QCP), and evaluate the dose–response function of the resulting RNA hairpins in a massively-parallel in vivo expression assay in bacteria. We train a convolutional neural network (CNN) on the OL sequences and their experimental binding scores, and use the resultant model to predict sequences that can bind the phage CPs with high affinity, which we then experimentally verify. Our results show that all three proteins occupy a predominantly unique sequence and structural binding space, thus providing a more complete picture of CP binding to RNA as compared with previous low-throughput binding assays. Finally, using the individual CP binding spaces, we were able to devise a design algorithm for customized non-repeat and orthogonal multi-binding-site RNA cassettes for CP-based applications in potentially any organism.

## Results

**Induction-based Sort-seq (iSort-seq).** We recently showed that placing a hairpin in the ribosomal initiation region of bacteria can lead to a ~×10–100 fold repression effect when bound to an RBP[23,32]. The magnitude of the effect allowed us to adapt this in vivo binding assay to a high-throughput OL experiment. We designed 10,000 mutated versions of the single wild-type (WT) binding sites of PCP, MCP, and QCP, and positioned each site at two positions within the ribosomal initiation region (Fig. 1a top and Supplementary Data 1). The library consists of three sub-libraries within the original library: binding sites that mostly resemble either the MS2-WT site, the PP7-WT site, or the Qβ-WT site (Fig. 1a bottom and Supplementary Fig. 1). We introduced semi-random mutations, both structure-altering and structure-preserving, as well as deliberate mutations at positions that previous studies have shown to be crucial for binding. Additionally, we incorporated into our library several dozens of control variants. We used variants characterized in our previous study as positive and negative controls[23,24,33,34] as follows: positive controls are binding sites that exhibited a strong fold-repression response, and negative control variants are either random sequences or hairpins which did not exhibit a fold-repression response. For the complete library, see Supplementary Data 1.

We incorporated each of the designed 10k single binding site variants downstream to an mCherry start codon (Fig. 1b) at each of the two positions (spacers $\delta = C$ or $\delta = GC$) to ensure high basal expression and enable detection of a down-regulatory response, resulting in 20k different OL variants. Each variant was ordered with five different barcodes, resulting in a total of 100k different OL sequences.

The second component of our system included a fusion of one of the three phage CPs to green fluorescent protein (GFP) (Fig. 1b) under the control of an inducible promoter. Thus, we created three libraries in *E. coli* cells; each with a different RBP but the same 100k binding site variants. In order to characterize

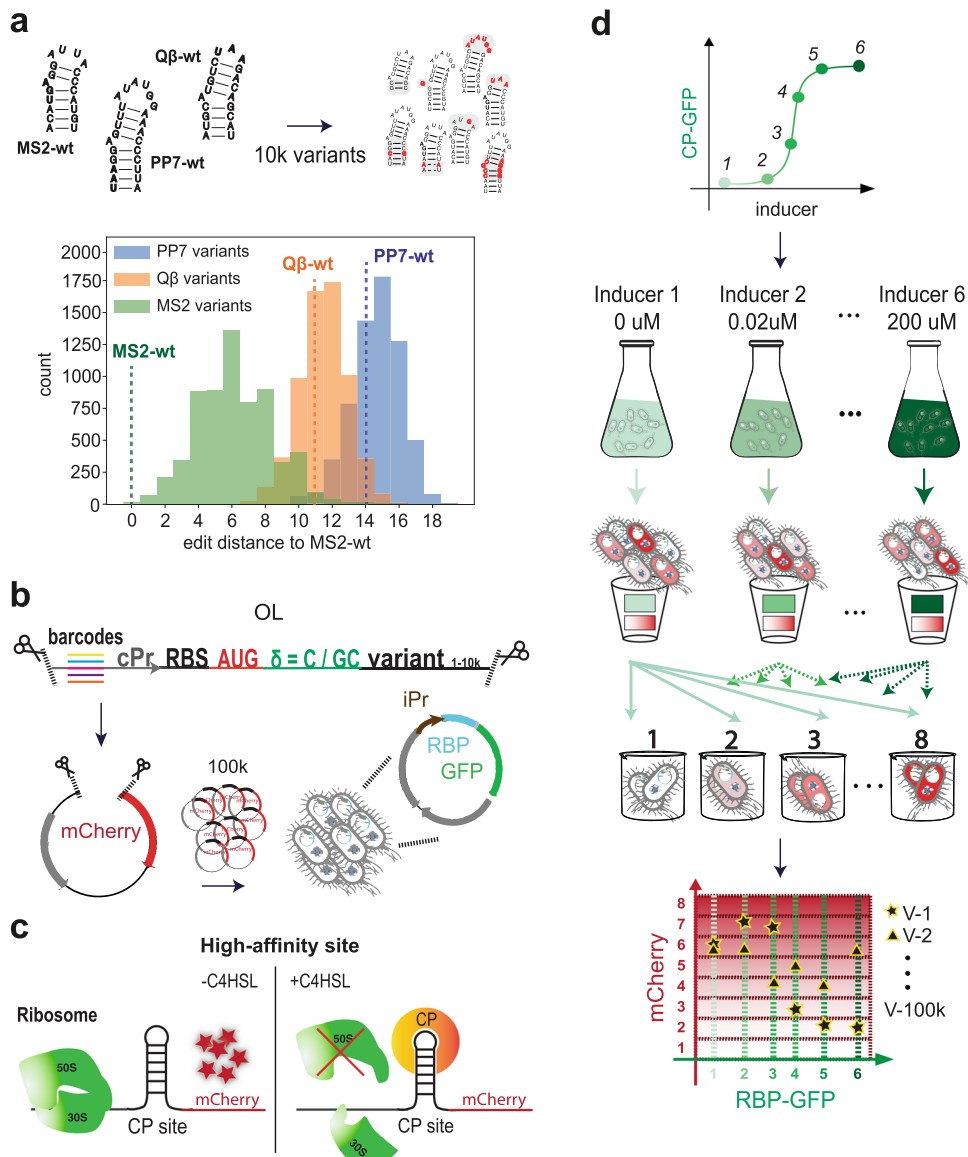

**Fig. 1 iSort-seq overview in *E. coli*. a** (Top) Wild-type binding sites for MS2, PP7, and Qβ phage coat proteins and illustrations of the 20k mutated variants created based on their sequences. (Bottom) Composition of the OL library. Histogram of the number of PP7-based variants (blue), Qβ-based variants (orange), and MS2-based variants (green) with different edit distances from the MS2-WT binding site. **b** Each putative binding site variant was encoded on a 210 bp oligo containing the following components: restriction site, barcode, constitutive promoter (cPr), ribosome binding site (RBS), mCherry start codon, one or two bases (denoted by δ), the sequence of the variant tested, and the second restriction site. Each configuration was encoded with five different barcodes, resulting in a total of 100k different OL variants. The OL was then cloned into a vector and transformed into an *E. coli* strain expressing one of three RBP–GFP fusions under an inducible promoter (iPr). The transformation was repeated for all three fusion proteins. **c** The schema illustrates the behavior of a high-affinity strain: when no inducer is added, mCherry is expressed at a certain basal level that depends on the mRNA structure and sequence. When inducer (C4-HSL) is added, the RBP binds the mRNA and blocks the ribosome from mCherry translation, resulting in a down-regulatory response as a function of inducer concentration. **d** The experimental flow for iSort-seq. Each library is grown at six different inducer concentrations, and sorted into eight bins with varying mCherry levels and constant RBP–GFP levels. This yields a 6 × 8 matrix of mCherry levels for each variant at each induction level. (Bottom) An illustration of the experimental output of a high-affinity strain (V1) and a no-affinity strain (V2). See Fig. S13 for flow cytometry gating strategy details.

the dose–response of our variants, each library was first separated into six exponentially expanding cultures grown in the presence of one of six inducer concentrations for RBP–GFP fusion induction. If the RBP was able to bind a particular variant, a strong fold-repression effect ensued, resulting in a reduced fluorescent expression profile (Fig. 1c). We sorted each inducer-concentration culture into eight predefined fluorescence bins, which resulted in a 6 × 8 fluorescence matrix for each variant, corresponding to its dose–response behavior. We call this

adaptation of Sort-seq "induction Sort-seq" (iSort-seq—for details, see Methods). As an example, we present a high-affinity, down-regulatory dose response for a positive variant (Fig. 1d, bottom V1), and a no-affinity variant exhibiting no apparent regulatory effect as a function of induction (Fig. 1d, bottom V2).

**Calculating binding scores.** We conducted preliminary analysis of the sequencing data to generate mCherry levels per RBP and inducer concentration for each variant (Fig. S2 and Methods). We

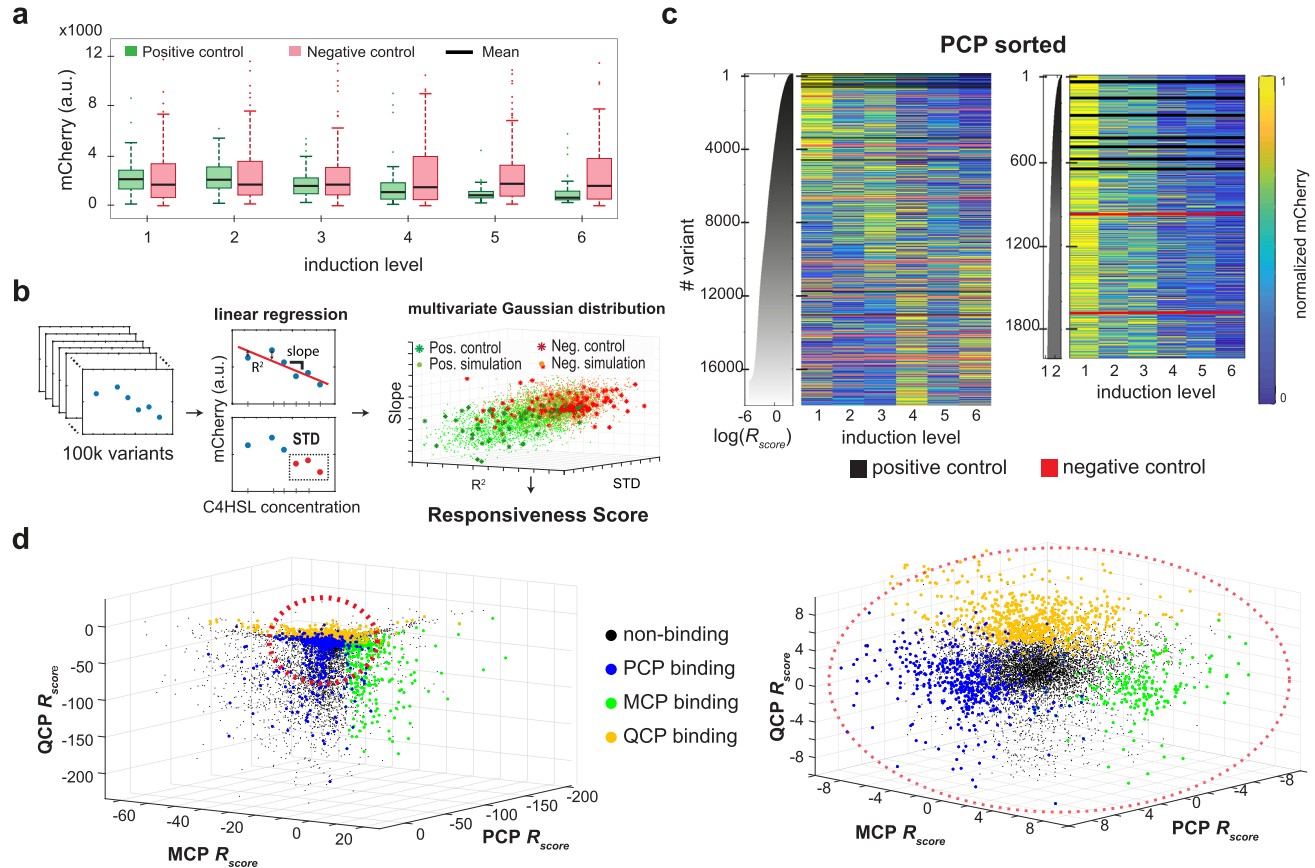

**Fig. 2 Responsiveness analysis and results. a** Boxplots of mCherry levels for the positive (green) and negative (red) control variants for each of the six induction levels for PCP-GFP, based on 75 different variants for positive control and 187 different variants for negative control. On each box, the central mark indicates the median, and the bottom and top edges of the box indicate the 25th and 75th percentiles, respectively. The value for 'Whisker' corresponds to approximately ±2.7 STD (standard deviation) and 99.3 percent coverage and extends to the adjacent value, which is the most extreme data value that is not an outlier. The outliers are plotted individually as plus signs. **b** Schema for responsiveness score ($R_{score}$) analysis. (Left and middle) Linear regression was conducted for each of the 100k variants, and two parameters were extracted: slope and goodness of fit ($R^2$). The third parameter is the standard deviation (STD) of the fluorescence values at the three highest induction levels. (Right) Location of the positive control (dark green stars) and negative control (red stars) in the 3D-space spanned by the three parameters. Both populations (positive and negative) were fitted to 3D-Gaussians, and simulated data points were sampled from their probability density functions (pdfs) (orange for negative and green for positive). Based on these pdfs the $R_{score}$ was calculated. **c** (Left) Heatmap of normalized mCherry expression for the ~20k variants with PCP. Variants are sorted by $R_{score}$. Black and red lines are positive and negative controls, respectively, and the gray graph is the $R_{score}$ as a function of variant. (Right) "Zoom-in" on the 2,000 top-$R_{score}$ binding sites for PCP. **d** (Left) 3D-representation of the $R_{score}$ for every binding site in the library and all RBPs. Responsive binding sites, i.e., sites with $R_{score} > 3.5$, are colored blue for PCP, green for MCP, and yellow for QCP. (Right) "Zoom-in" on the central highly concentrated region. Source data are provided as a Source data file. Altogether, we identified 1868, 1144, and 2624 binding sites (i.e., $R_{score} > 3.5$) for PCP, MCP, and QCP, respectively. In addition, there were an additional 3736, 1460, and 4682 "non-classified" binding sites (i.e., $0 < R_{score} < 3.5$) for PCP, MCP, and QCP, respectively, while the rest were determined to be non-binding ($R_{score} < 0$).

also eliminated variants for which we acquired too few reads (see Methods for additional details). To ascertain the validity of our assay, we first characterized the behavior of our control variants (Fig. 2a). A linear-like down-regulatory effect as a function of RBP induction is observed for the positive control variants (green), while no response in mCherry levels is observed for the negative controls (red). Additionally, the spread in mCherry at high induction levels is significantly smaller for the positive control variants than that of the negative control variants.

Next, to sort the variants in accordance with their likelihood of binding the RBP (i.e., similarity of their dose–response to the positive controls), we carried out the following computation (for details, see Methods). First, we characterized all variants by calculating a vector composed of three components: the slope of a linear regression, its goodness of fit ($R^2$), and standard deviation of the fluorescence value at the three highest induction bins (Fig. 2b, middle). Next, we computed two multivariate Gaussian

distributions using the empirical 3-component vectors, that were extracted for the positive and negative controls and for the given RBP, to yield a probability distribution function (pdf) for both the responsive and non-responsive variants, respectively (Fig. 2b, right). The two populations are relatively well-separated from one another, presenting two distinct clusters with minor overlap. Finally, we defined the "responsiveness score" for each variant ($R_{score}$ - see Methods for a formal definition) as the logarithm of the ratio of the probabilities computed by the responsive pdf to the non-responsive pdf. This score was computed for each unique barcode, and the final result for a sequence variant was averaged over up to five vectors, one for every variant barcode that passed the read-number and basal-level thresholds (see Supplementary Fig. 2 and Methods).

In Fig. 2c left, we plot the expression heatmap of the ~18k variants with PCP sorted (top to bottom) by decreasing $R_{score}$ (see also Fig. S3 for MCP and QCP). The plot shows that 5470

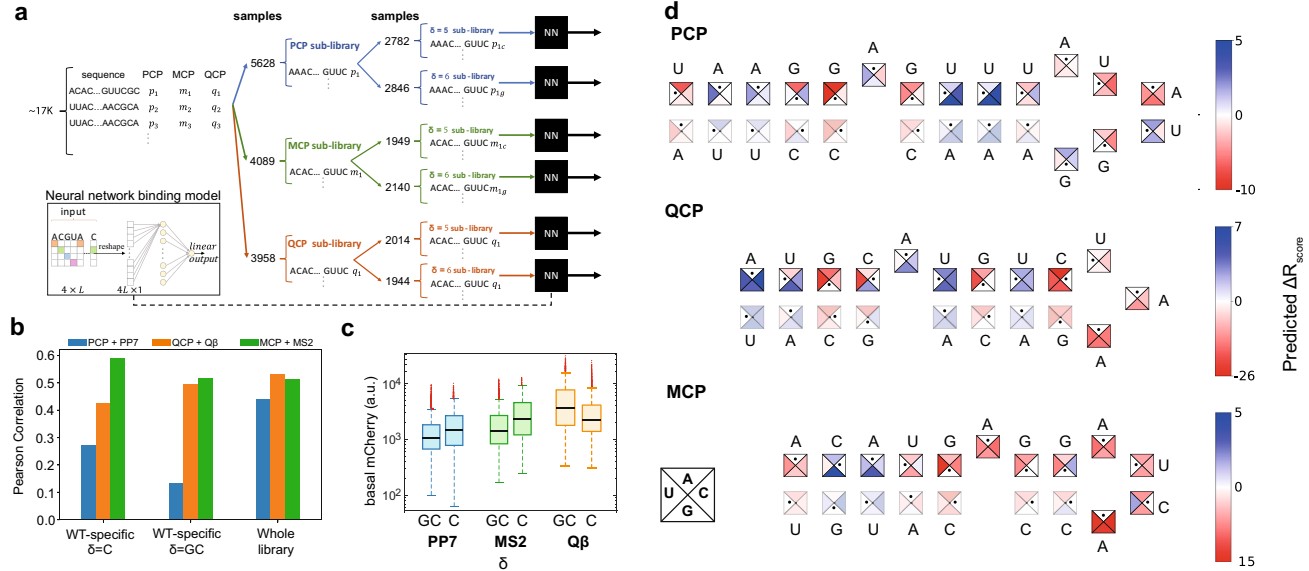

**Fig. 3 Analysis of MCP, PCP, and QCP RNA-binding sequence preferences. a** Scheme for the data preparation and neural network architecture (inset) used. **b** Pearson correlation over a held-out test set computed for the WT-specific sub-libraries (i.e., PCP, MCP, and QCP with PP7-based, MS2-based, and Qβ-based binding sites, respectively at either δ = C (left) or δ = GC (middle)), and for the whole-library CNN model (right). **c** Boxplots of basal mCherry levels for the six WT-specific sub-libraries, based on the following number of strains (from left to right): 3113 (PP7-GC), 3067 (PP7-C), 2810 (MS2-GC), 2743 (MS2-C), 2702 (Qβ-GC), 2743 (Qβ-C). On each box, the central mark indicates the median, and the bottom and top edges of the box indicate the 25th and 75th percentiles, respectively. The value for 'Whisker' corresponds to approximately ±2.7 STD (standard deviation) and 99.3 percent coverage and extends to the adjacent value, which is the most extreme data value that is not an outlier. The outliers are plotted individually as plus signs. **d** Illustrations of the whole-library model predictions for the three sub-libraries for any single- or double-nucleotide structure-preserving mutation. Each binding site is shown, with the wild-type sequence indicated as black dots inside the squares. Each square is divided to the four possible options of nucleotide identity, with the colors representing the predicted change in $R_{score}$ with respect to the wild type for each option. Source data are provided as a Source data file.

variants exhibit an apparent down-regulatory response, defined as $R_{score} > 0$, corresponding to having a larger probability of belonging to the positive control distribution as compared with the negative control distribution. By comparison (Supplementary Fig. 3), MCP and QCP yielded 2604 and 7306 such variants, respectively. This indicates that while QCP may be the most promiscuous RBP in our library (i.e., tolerates a more varied set of binding sites), MCP is likely to be the most limited in terms of binding specificity. A closer observation of the top of the list (top 2000, Fig. 2c, right) indicates that for a high $R_{score}$, a rapid reduction in fluorescence is detected in the second bin, which indicates that these variants also seem to exhibit the strongest binding affinity. Sorted $R_{score}$ values for each RBP as well as the ΔΔ$G$ values derived from those scores (see Supplementary Fig. 3 and Methods) are available in Supplementary Data 1. We next plot the $R_{score}$ obtained for all three RBPs for each variant (Fig. 2d). We overlay the plot with colored dots corresponding to the variants with $R_{score} > 3.5$ in each list, which are the most specific variants. The plots reveal very little overlap between the subsets of variants that are highly responsive to the different RBPs, indicating that the vast majority of these highly-responsive binding sites are orthogonal (i.e., respond to only one RBP), which was expected for PCP and MCP and PCP and QCP, but not necessarily for MCP and QCP whose WT sites are not mutually orthogonal[23–25,28,35,36].

**RNA-binding sequence preferences**. Using empirical $R_{score}$ values and associated binding site sequences as a training set, we developed an ML-based method that predicts the $R_{score}$ values for every mutation in the WT sequences. We first built a model

specific to each protein and its WT binding site length. To do so, we used a neural network that receives as input the sequence of a binding site the same length as the WT sequences (25 nt for PP7-WT, 19 nt for MS2-WT, and 20 nt for Qβ-WT) and outputs a single score. We trained a specific network for each of the three RBP-OL experiments and the two positions where the binding sites were embedded within the ribosomal initiation region (Fig. 3a and Supplementary Fig. 4), resulting in a total of six different models. Such a model preserves the positional information for each feature, i.e., the position of each nucleotide in the WT binding site. To choose the prefix (δ) in which more robust scores were measured, we looked at the Pearson correlation over a held-out test set. The correlations for the most robust position yielded values of 0.27 for PCP with PCP-based sites and δ = C, 0.59 for MCP with MCP-based sites and δ = C, and 0.5 for QCP with QCP-based sites and δ = GC (Fig. 3b). Interestingly, the variant group with higher Pearson correlation was also characterized by higher basal mCherry expression levels (Fig. 3c), which in turn resulted in a higher fold repression effect. Thus, higher correlation, meaning more robust predictability, correlated with higher fold-repression, which provided additional validity to our analysis.

In order to better understand the relationship between binding site sequence and binding, we developed a protein-specific model based on the whole library, which we termed whole-library model. This model, as opposed to our WT-specific model, enables binding prediction to any site (i.e., of length different than the WT-site length). The model is based on a CNN and receives as input nearly all of the oligo library sequences (~17,000). As with the protein-specific NN-model, we looked at the Pearson correlation over a held-out test set (Fig. 3b, right) with the

CNN model and found a significant improvement in Pearson correlation for PCP, while the correlation for MCP and QCP remained approximately the same.

We used the whole-library model to analyze the effect of structure-conserving mutations in each of the WT binding site sequences (Fig. 3d). We present the ML model's results as "binding rules" depicted in illustrations for each of the three CP binding sites. The schemas represent the predicted change in responsiveness with respect to the WT sequence for every single-nucleotide mutation (SNP) in the loop or the bulge region, and every di-nucleotide mutation (DNP) preserving stem structure in the stem regions. For instance, in the schema for PCP (Fig. 3d, top), mutating the bulge from A to C reduces the binding site's predicted responsiveness. By contrast, mutating the top base-pair in the upper stem from a U–A to a C–G, and the bulge from an A to a U/G are both predicted to increase the responsiveness score with respect to the WT binding site. A clear characteristic of PCP is the tolerance to DNPs in the stem regions, which is reflected by the dominance of the blue colors or light red (indicating a small reduction in responsiveness with respect to the WT binding sites), while there are only a few bases where single mutations are found to abolish binding (e.g., UA portion of the loop). It is important to note that our results for PCP broadly correlate with past works[23,24,26], which found the loop and the bulge regions to be critical for PCP binding, while sequence variations in the stems did not alter binding significantly. For QCP (Fig. 3d, middle), a significantly different picture emerges. Our results indicate that the WT sequence we used, as referred to in the literature[22,23,25], has a lower $R_{score}$ than many mutated versions of it. The bulge, for instance, has a higher $R_{score}$ with C/G instead of the WT A. The data seems to indicate that QCP prefers a four nucleotide B-rich (i.e., C/G/U) stem and a short C/G bulge or loop mini-motif. This motif is apparent throughout the binding site, as can be seen from the blue-colored nucleotides of both the lower and upper stems. For MCP (Fig. 3d, bottom), a tolerance to DNPs in the lower stem emerges from our analysis, while a strong sensitivity to SNPs in the bulge, upper stem, and the loop regions is revealed. Past analysis[23,27,34] also highlighted the sensitivity to mutations in the loop and the bulge regions, indicating that the in vivo environment does not alter the overall binding characteristics of MCP.

Finally, to provide a sanity check on our structural findings, we reanalyzed the raw iSort-seq data using an average nearest neighbor (ANN) approach (see Methods) to calculate a non-parametrized $R_{score}$. We first computed the cross-correlation between the non-parametrized and the Gaussian-parametrized $R_{score}$ (Supplementary Fig. 5), and obtained a Pearson correlation coefficient of ~0.5 between both sets of scores for all three proteins. We then retrained the whole-library CNN model using the non-parametrized scores, and obtained Pearson correlation values on a held-out test set of 0.53, 0.41, and 0.38 for PCP, MCP, and QCP as compared with 0.44, 0.51, and 0.53, respectively, with the Gaussian-parametrized $R_{score}$. Next, we recomputed the binding preferences visualized on the structures as shown in Fig. 3d (see Supplementary Fig. 6). The figure shows that while there is some deviation between the structures obtained from both models, most trends are nevertheless sustained.

**RNA-binding structure preferences**. To better understand the relationship between binding site structure and binding, we extended the CNN model to include structural information (Fig. 4a). This model, as opposed to our whole-library model, incorporates both the sequence and secondary structure of the RNA binding site, as calculated by RNAfold[37]. All three CNNs showed improved predictive performance when the structural data was added into the network (Supplementary Fig. 7).

We used this model to analyze the effect of structure-altering mutations on protein binding. To do so, we generated various binding sites with a predefined structure and used the whole-library models to predict their responsiveness score. Specifically, we looked at three types of mutations: alteration of upper-stem length, alteration of loop length, and alteration of bulge size. Overall, upper-stem length plays a big role in binding affinity for all three RBPs, though not equally (Fig. 4b, left). PCP seems to be the most resilient to both shorter and longer upper-stems, while MCP does not seem to tolerate any upper-stem length which deviates from the canonical 2 bp. Finally, QCP exhibits some tolerance to alternative stem lengths, especially by an increase of 1 bp.

Varying the loop-length suggests increased flexibility for all three RBPs (Fig. 4b, right). PCP is the most resilient, displaying a viable binding affinity to loops that range from five to eight nucleotides in length. MCP is less tolerant, displaying flexibility only for structures containing loops that are five nucleotides in length. Finally, QCP shows some tolerance to longer loop lengths, especially for five nucleotides in length.

Finally, examining the importance of the bulge, a high variation in tolerance to mutations for the three CPs is observed (Fig. 4c). PCP can tolerate and even have higher affinity with sequences that either have no bulge, or a two-nucleotide bulge. This is depicted by a non-negligible variant density above the 3.5 threshold, and indicates that a bulge is not necessarily needed for binding. Together, with the SNP analysis depicted in Fig. 3d, it would seem that PCP can bind structures with a predominantly G-rich single stem of varying length and a 5–8 nucleotide loop containing a 3–4 nucleotide U/A segment. MCP, on the other hand, has negligible tolerance for variants with no bulge, and very low tolerance for those with a two-nucleotide bulge. This sensitivity correlates with MCP's previous structure and sequence dependencies of the loop and upper stem (Figs. 3d and 4b). QCP displays some tolerance to both bulge mutations, though much less than PCP, and thus seem to tolerate some deviation from the consensus bulge-stem-loop structure.

In summary, the structural analysis indicates that all three proteins prefer different structures, with some overlap that can create cross-binding. This is consistent with QCP's known[23,25,33] weak binding affinity to the MS2-WT binding site, and MCP's ability to bind Qβ-WT. PCP seems to prefer a structure with an upper stem of length four base-pairs or longer, a variable loop size ranging from five to eight nucleotides with some sequence specificity, and a weak dependence on bulge. MCP is constrained in both structure and sequence specificity needing a bulge separating a lower and upper stem, two base-pair upper stem, and a loop length of four to five nucleotides in length with a conserved sequence signature. Finally, QCP seems to display a binding signature consistent with a repeat concatemer of 5-B-rich-stem-bulge sequence and structural motif.

**Validations—new cassettes for RNA imaging**. To validate both our experimental measurements and model predictions, we compared our results to a previous study that measured high-throughput in vitro RNA-binding of MCP[27]. In the study, the researchers employed a combined high-throughput sequencing and single-molecule approach to quantitatively measure binding affinities and dissociation constants of MCP to more than $10^7$ RNA sites using a flow-cell and in vitro transcription. The study reported $\Delta G$ values for over 120k variants, which formed a rich dataset to test correlation with our measured and predicted $R_{score}$ values. First, we computed Pearson correlation coefficient of the purely experimental measurements for variants that were both in our library and in the in vitro study. The result (Fig. 5a, left)

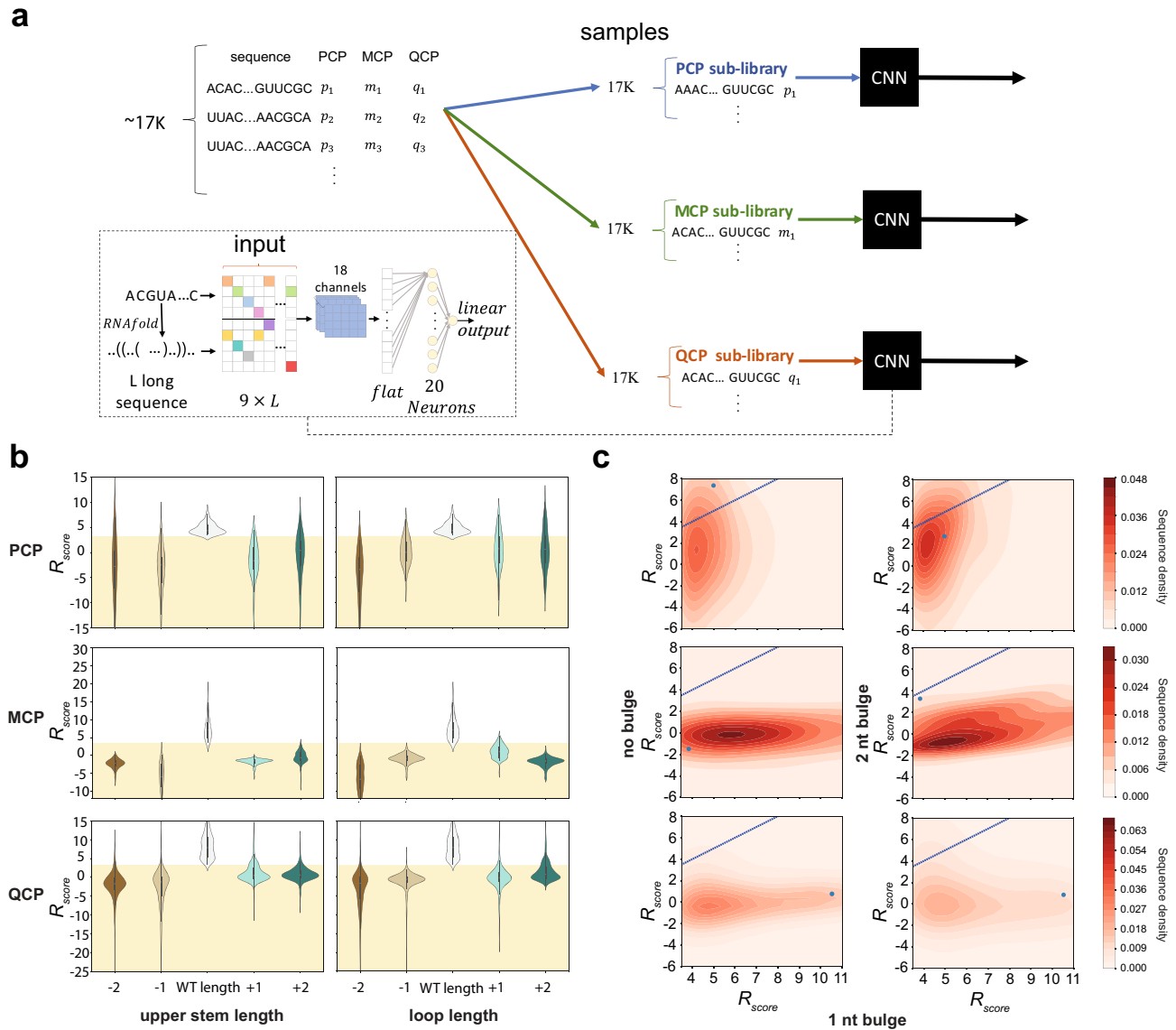

**Fig. 4 Analysis of MCP, PCP, and QCP RNA-binding structure preferences. a** A scheme for the data preparation and neural network architecture (inset) used for the protein-specific CNN model based on the whole-library. We generated various binding sites with a predefined structure different from the wild type and used the whole-library models to predict their responsiveness score. **b** Predicted $R_{score}$ distributions for binding sites that differ in the length of the upper stem (left) or the loop (right) for PCP (top row), MCP (middle row), and QCP (bottom row). Stem and loop lengths were varied by ±2 base-pairs and nucleotides, respectively. Yellow shade corresponds to non-binding $R_{score}$ values. **c** Density maps for predicted $R_{score}$ for either no bulge (left-column) or a 2-nucleotide bulge (right-column) mutation of a wild-type-like structure for PCP-response (top row), MCP-response (middle row), and QCP-response (bottom row). Source data are provided as a Source data file. Blue dots correspond to wild-type sequences.

indicates a positive and statistically significant correlation ($r = 0.23$). We next predicted $R_{score}$ values using the whole-library model for all the reported variants of the in vitro study (Fig. 5a left-to-right), and found a strong correlation ($r = 0.44$) for single-mutations variants, a moderate correlation ($r = 0.4$) for double-mutation variants and a weak correlation ($r = 0.17$) with the entire set of 129,248 mutated variants. Given the large difference between the experiments and the different sets of variants used (e.g., in vitro vs. in vivo, microscopy-based vs. flow cytometry-based), the positive correlation coefficients ($p$-values < 0.0007 for all reported coefficients) indicate a good agreement for both sets of experimental data, and a wide applicability for the learned binding models for MCP.

To examine the wider applicability of the findings, we generated cassettes containing multiple non-repetitive CP binding sites identified by our experimental dataset, and tested them in

both bacterial and mammalian cells. Once labeled with a fusion of the CP to a fluorescent protein, functional cassettes appear as trackable bright fluorescent *foci* (Fig. 5b). Initially, we designed two binding site cassettes based on library variants that were identified as highly responsive for each CP and tested them in bacteria: the first with four PP7-WT and five Qβ-WT binding sites interlaced, and the second with ten QCP binding sites identified as strongly bound in our library. The cassettes were cloned downstream of a pT7 promoter on a single copy plasmid and transformed to *E. coli* BL21 cells (Fig. 5b, top), together with a plasmid encoding the corresponding CP (QCP, PCP), which was fused to mCherry. In addition, we also cloned the state-of-the-art PP7-24x repeat cassette, which has been used in many studies[38] to provide a benchmark by which to assess our results. For all cassettes tested, several hours after induction of the T7 RNAP, bright spots began to form in the polar regions of the

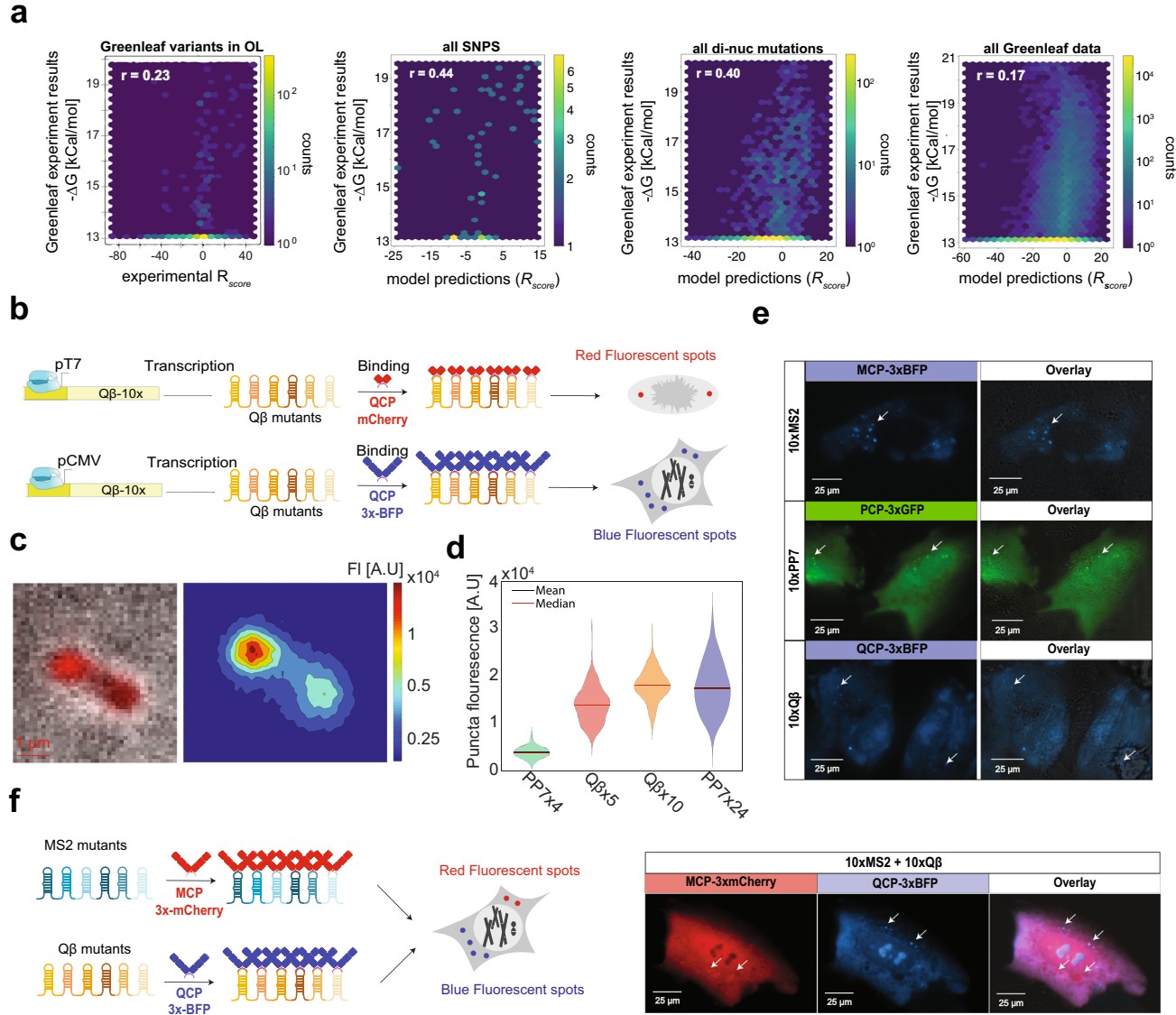

**Fig. 5 Validations: cassettes for RNA imaging in U2OS cells. a** $R_{score}$ comparison to $\Delta G$ results of a previous study that reported MCP binding to more than 129k sequences[27]. Each plot (from left to right) represents Pearson correlation coefficient using: the experimental measurements for variants that were both in our OL and in the in vitro study, the $R_{score}$ values predicted by our ML model for all single-mutation variants, for all double-mutation variants, and for the entire set of 129,248 mutated variants. **b** Example experiment design for bacterial and mammalian cells. High $R_{score}$ binding sites were incorporated into a ten-site cassette downstream to either a T7 promoter in *E. coli* or a mammalian CMV promoter. When the matching QCP-mCherry for bacteria (top) or QCP-3xBFP for mammalian cells (bottom) are added, it binds the binding site cassette and creates a fluorescent spot. **c** Fluorescent image of typical bacterial cell depicting bright puncta in cell pole. (Left) Raw image. (Right) Processed image depicting variations in brightness density. **d** Violin plots of mean spot intensity measured in bacteria expressing the different binding sites cassettes. Values are based on the following number of captured cells taken on multiple days (from left to right): 99 (PP7-4x), 149 (Qβ-5x), 187 (Qβ-10x), 182 (PP7-24x). **e** The results for three CP-specific cassettes transfected with the matching CP-3xFP plasmid into U2OS cells and imaged by fluorescence microscopy for detection of fluorescent *foci*. For each experiment, both the relevant fluorescent channel and the merged images with the differential interference contrast (DIC) channel are presented. Each experiment was successfully conducted in duplicates and on two different days. **f** (Left) Experimental design for the orthogonality experiment: two separate cassettes with 10 predicted mutated sites for either MCP only or QCP only, respectively, were designed and transfected together with both MCP-3xmCherry and QCP-3xBFP, into U2OS cells. (Right) Results for the orthogonality experiment: a cell presenting non-overlapping fluorescent *foci* from both fluorescent channels, indicating binding of MCP and QCP to different targets. This experiment was successfully conducted in duplicate and on two different days. Fluorescent wavelengths used in these experiments are: 400 nm for BFP, 490 nm for GFP, and 585 nm for mCherry. Source data are provided as a Source data file.

cells; these spots were 2–3 times brighter than the cell background fluorescence (Fig. 5c). We then measured the mean intensity of the spots (~100–200 cells for each cassette type). The intensity distributions (Fig. 5d) show a dependence on cassette size. Surprisingly, the mean brightness for the Qβx10 cassette was slightly higher than the mean brightness measured for the PP7x24

despite having less than half the binding sites. In addition, the mean brightness for the Qβx5 cassette was estimated at ~70% of the value measured for PP7-24x. Furthermore, the mean intensity distributions obtained for both the Qβx5 and Qβx10 cassettes were narrower than the one obtained for the PP7-x24 cassette. The increased brightness per binding site of the new cassettes and

the narrower distributions are likely related to the sequence instability of the PP7-24x in vivo, which is encoded by a repeat DNA sequence.

Next, we designed additional cassettes for validation in mammalian cells. Each cassette was designed with ten different binding sites, all characterized by a large edit distance (i.e., at least 5) from the respective WT site and from each other, thus creating a sufficiently non-repeating cassette that Integrated DNA Technologies, Inc. (IDT) was able to synthesize in three working days. In addition, all selected binding sites exhibited non-responsive behavior to the two other CPs in our experiment. We cloned the cassettes into a vector downstream to a CMV promoter for mammalian expression and transfected them into U2OS cells together with one of the RBP-3xFP plasmids encoding either PCP-3xGFP, MCP-3xBFP, or QCP-3xBFP. In a typical cell (Fig. 5e), all three cassettes generated more than five fluorescent puncta, dispersed throughout the cytoplasm. The puncta were characterized by rapid mobility within the cytoplasm, and a lack of overlap with static granules or distinct features which also appear in the DIC channel (see Supplementary movies 1–3). Negative control experiments, where CP-3xFP plasmids were transfected with either an empty plasmid (pUC19) or non-cognate binding site cassettes, did not show such puncta (Fig. S8 and Supplementary Movie 6).

To expand our claim to orthogonal and simultaneous imaging of multiple promoters, we ordered two additional cassettes encoded with MS2 and Qβ variants, respectively, and co-transfected them with a plasmid encoding both of the matching fusion proteins: MCP-3xmCherry and QCP-3xBFP (Fig. 5f, left). For each cassette, the sites were chosen with two constraints: to minimize repeat sequences, and to maximize orthogonality to the other CPs (e.g., both MS2-WT and Qβ-WT binding sites were not included as they exhibit cross-responsiveness and are thus not orthogonal). In Fig. 5f, right we plot sample cell images depicting single and double channel views. The images show that both cassettes produce a spatially distinct set of puncta (Fig. 5f, left and middle), which can be definitively associated with one of the two proteins (Fig. 5f, right). This indicates that our binding sites are sufficiently orthogonal to allow the tracking of more than one cassette simultaneously. Moreover, there is little difference between the number of puncta of the two sequences, and the fluorescent intensity for all puncta seem to fluctuate unimpeded in all three directions ($x$, $y$, and $z$) inside the cell. Taken together, the microscopy experiments conducted in bacterial and mammalian cells demonstrate the universal applicability of the results obtained from the high-responsiveness binding sites identified in the OL experiment to the advancement of RNA imaging in a variety of cell types.

**De novo design of dual-binding site cassettes**. Finally, we wished to validate the model's predictive capability by creating cassettes with binding sites that did not exist in our experimental library. We used the whole-library models to predict de novo functional binding site sequences, which could bind multiple CPs. To do so, we generated all possible variants with Hamming distance 3–7 to one of the three WTs. From this set of sequences, we randomly selected one million sequences and used the models to predict the responsiveness score for each of the three CPs. In Fig. 6a, we plot the variant density distribution based on predicted $R_{score}$ values. The plots show that the highest density of sequences appears at $R_{score}$ values that hover around 0 for all three proteins. The plots further show that there is a bias towards negative responsiveness values for all three proteins in the computed sequences. This is consistent with having a small region of sequence space which facilitates specific binding, which, in turn,

is easy to abolish with a small number of mutations. In contrast, high responsiveness scores are only computed for a small number of the sequences, as can be seen by the sharp gradient in the density plot for positive responsiveness values. Finally, each plot shows a low-density region containing sequences that exhibit a high responsiveness score for both CPs. These sequences are predicted to be dual binders.

To test the predictions of the whole-library models experimentally, we designed another 10× binding site cassette (Fig. 6b), where each binding site was selected from the set of predicted sequences whose responsiveness scores for QCP and PCP were both above 3.5 (see dashed square in Fig. 6a, left panel). Therefore, we expected the cassette to generate fluorescent *foci* when bound by either QCP or PCP. As before, we cloned the cassette into a vector downstream of a CMV promoter for mammalian expression and transfected it into U2OS cells together with a plasmid encoding either PCP-3xGFP or QCP-3xBFP. In Fig. 6c, we plot fluorescent and DIC images for PCP (left) and QCP (right), depicting bright fluorescent *foci* that are located outside of the nucleus and which do not overlap with a DIC feature. The plots show distinct puncta observed with both relevant RBPs, confirming the dual binding nature of the cassette. An additional cassette containing predicted PP7 sites also presented mobile fluorescent *foci* when tested in a similar manner with PCP-3xGFP (Supplementary Fig. 9 and Supplementary Movie 7). Consequently, these images support our model's ability to accurately predict MCP, PCP, and QCP binding sequences with known function with respect to all three RBPs.

## Discussion

In this study, we adapted our previously developed binding assay for quantifying CP binding site affinity in vivo to a high-throughput OL platform in bacteria[23,32]. We were inspired by the need to develop approaches for understanding RNA–protein interactions. It is generally believed that the combinatorial nature of RNA sequence and its intramolecular interactions lead to high complexity, making simulations based on biophysical models a difficult task with limited degree of success[37,39–41]. As a result, little is known about the evolutionary constraints on RNA structures, making bioinformatic identification of functional RNAs difficult[27]. Moreover, recent advances in OL analysis have shown that by generating a sufficiently large experimental dataset, and subsequently using it to train a ML model, biological insights into transcriptional regulation can be obtained[10–12]. Given these recent advances, we hypothesized that an OL-ML approach could generate insights into in vivo characterization of protein–RNA binding as well. We chose to test our hypothesis on phage CPs, which on the one hand corresponds to a classic model system for RNA–protein interactions, while on the other hand constitutes an important biomolecular tool for various biological imaging applications.

Using the OL-ML approach, we experimentally identified several thousand functional binding sites for the phage coat proteins of PP7, MS2, and Qβ. This dataset, in turn, was used to train a ML model, which allowed us to quantitatively assess the in vivo sequence and structural binding space of the three phage CPs to RNA. Based on the model and data, we found that each CP occupies a different sequence and structural space, with some overlap. Using this broader binding characterization, we concluded that the WT binding site for QCP is sub-optimal, and that dual-binding binding sites such as Qβ-WT (QCP and MCP) or MS2-WT (MCP and QCP) are the exception rather than the rule, which in turn provide the community with a third orthogonal CP channel (QCP) for various imaging or genome-editing applications. Furthermore, we demonstrated that the dataset of single

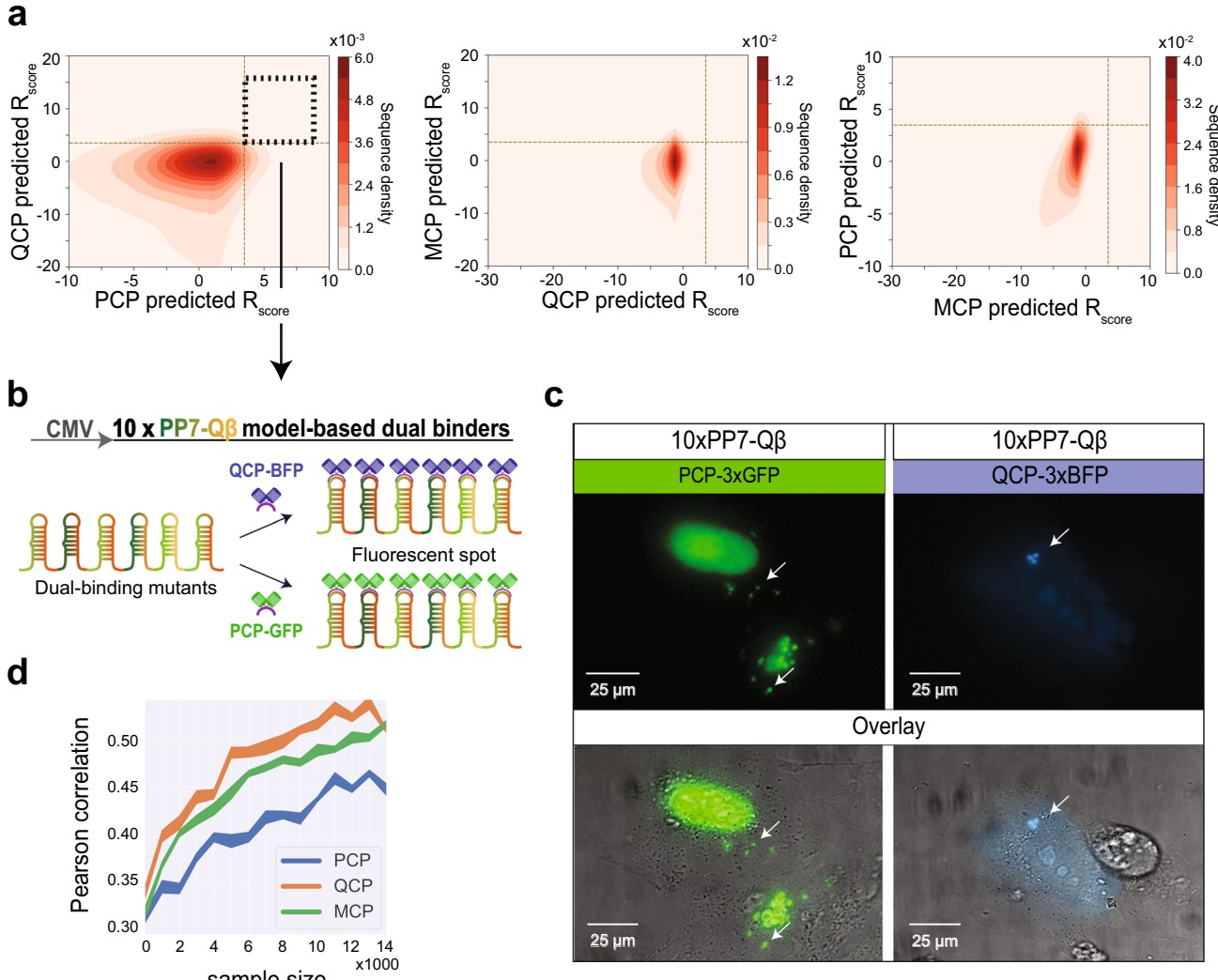

**Fig. 6 De novo design of dual-binding site cassettes in U2OS cells. a** 2D density plots (pink-red scale) depicting the predicted $R_{score}$ values for one million ML variants binding to (left-to-right): PCP and QCP, MCP and QCP, and MCP and PCP. QCP-PCP dual-binding variants are located in the black dashed square. **b** Based on the dual-binding mutants for QCP and PCP from our model predictions, we designed an additional cassette. **c** Results for the dual-binding experiment. Fluorescent *foci* can be observed for the cassette expressed with either PCP-3xGFP or QCP-3xBFP. For both experiments, both the relevant fluorescent channel and the merged images with the DIC channel are presented. Each of the two experiments were successfully conducted in duplicates and on two different days. Fluorescent wavelengths used in these experiments are: 400 nm for BFP and 490 nm for GFP. **d** Evaluation of prediction accuracy based on size of the training set. For each training set size, a random set of more than 1,000 training-set variants was withheld for computational testing post-training. Performance is reported as average Pearson correlation over 10 random training and test sets (and standard deviation in shade). Source data are provided as a Source data file.

RNA binding sites obtained in bacteria is sufficient for training a model, which can generate synthetic long non-coding RNAs (slncRNA), that were functional in mammalian cells. This is evidence that, at least for this set of proteins, the RNA-RBP module can accommodate multiple cellular environments, thus constraining the complexity of the overall system. Consequently, our work paves the way for characterizing and predicting the binding of additional RBPs, in addition to validating the utility of the OL-ML approach to RNA-related problems.

In addition to providing a deeper characterization of the sequence and structural determinants of CP binding to RNA, we were also inspired by the need to solve an important technological bottleneck that has hindered progress in the use of CP-based synthetic constructs for various questions in biology. As a result of this work, we now provide the community with a computational tool that allows for rapid design, synthesis, and evaluation of CP-binding slncRNA molecules that do not contain sequence

repeats. The elimination of sequence repeats will also remove many of the previous restrictions associated with these systems, such as the need for repetitive cloning cycles, repeat-based structure formation, and limitation on the number of functional binding sites. In total, these achievements provide the community with a reliable design tool that can integrate both experimentally verified and predicted binding sites for phage CP binding cassettes in a variety of organisms.

Finally, together with other recent studies[10–13], our work provides another example for the potential utility of the OL-ML approach for deciphering biomolecular mechanisms. In our case, several thousand variants generated a Pearson correlation of ~0.5, which is a poorer result when comparing with other recent OL-ML studies (Pearson correlation ~0.8). However, despite this result, we were still able to generate a reliable predictive algorithm for CP-binding slncRNA design. Moreover, correlations obtained by our model on an independent dataset of in vitro measurement

of MCP were also in the range of ~0.5. On the computational side, we found that searching hyper-parameters in cross-validation (3 folds in our case, as in DeepBind), and testing tens of random sets (25 in our case) optimized performance in finding a robust set of hyper-parameters that were neither over-fitted to one fold, nor too far from the optimal hyper-parameters set. Thus, a Pearson correlation of 0.5 seems to be a sufficiently good indicator, at least for our system. The discrepancy in Pearson correlation between our work and the previous studies can be attributed to two major differences. The first is likely due to having only a limited set (several thousand per CP) of positive variants by which to train the model. Indeed, more data is expected to increase the predictive power of our models (Fig. 6d). The second is rooted in the fact that using different ML strategies and introducing RNA secondary structural interactions (Supplementary Fig. 7) did not yield an improvement in the Pearson correlation beyond this range. This may be indicative of an insufficient understanding of the underlying thermodynamic constraints, which guide RNA folding and its interaction with CPs. Consequently, at the present time, it is impossible to tell whether a Pearson correlation of 0.5 is typical or "surprisingly" small for RNA systems. Future work on other complex RNA-based molecular interaction systems will determine whether the OL-ML approach will indeed prove to be a useful tool for providing both mechanistic and structural insights, and a source of data for formulating reliable design algorithms in synthetic biology.

## Methods

### Construction of the oligo library
We designed 10,000 mutated versions of the WT binding sites of the phage CPs of PP7 (Fig. 1 and Fig. S1), MS2 and Qβ, and positioned them at two positions within the ribosomal initiation region. Each of the designed 10k sites was positioned either one or two nucleotides downstream to the mCherry start codon, resulting in 20k different configurations. We then ordered the following OL from Agilent: 100k oligos (Supplementary Data 1), each 210 bp long, containing the following components: BamHI restriction site, barcode (five for each variant, see Supplementary Data 5), constitutive promoter (cPr), ribosome binding site (RBS), mCherry start codon, one or two bases (denoted by δ), the variant binding site, ~60 bp of the mCherry gene, and an ApaLI restriction site. We then cloned the OL using restriction-based cloning strategy. Briefly, the 100k-variant ssDNA library from Agilent was amplified in a 96-well plate using PCR (see Supplementary Data 2 for primers), purified, and merged into one tube. Following purification, dsDNA was cut using BamHI-HF and ApaLI (New England Biolabs [NEB]) and cleaned. Resulting DNA fragments were ligated to the target plasmid containing an mCherry open reading frame and a terminator, and a kanamycin (Kan) resistance gene, using a 1:1 ratio. Ligated plasmids were transformed to E. cloni® cells (Lucigen) and plated on 37 large agar plates with Kan in order to conserve library complexity. Approximately two million colonies were scraped and transferred to an Erlenmeyer for growth. After O/N growth, plasmids were extracted using a maxiprep kit (Agilent), their concentration was measured, and they were stored in a microcentrifuge tube in −20 °C.

### Construction of RBP–GFP fusions
RBP sequences lacking a stop codon were amplified via PCR off either Addgene or custom-ordered templates (Genescript or IDT, see Supplementary Data 3). MCP, PCP, and QCP were cloned into the RBP plasmid between restriction sites KpnI and AgeI, immediately upstream of a GFP gene lacking a start codon, under the pRhlR promoter (containing the rhlAB las box38) and induced by N-butanoyl-L-homoserine Lactone (C4-HSL, Cayman Chemical). The backbone contained an ampicillin (Amp) resistance gene. The resulting fusion-RBP plasmids were transformed into E. coli Top10 cells. After Sanger sequencing, positive transformants were made chemically competent and stored at −80 °C in 96-well format.

### Double transformation of OL and RBP–GFP plasmids
Note: the following two sections were conducted three times, one for each of the RBP–GFP fusions.

OL DNA was transformed into ~300 chemically competent bacterial cell in 100 μl aliquots containing one of the RBP-GFP plasmids in 96-well format. After transformation, cells were grown in 2 L liquid Luria Broth (LB) with twice the concentration of the antibiotics—Kan and Amp—overnight at 37 °C and 250 rpm. After growth glycerol stocks were made by centrifugation, re-suspension in 30 ml LB, mix 1.2 ml with 400 μl 80% glycerol—20% LB solution and stored in −80 °C.

### Induction-based Sort-seq OL assay
One full glycerol stock of the library was dissolved in 500 ml of LB with Amp and Kan and grown overnight at 37 °C and 250 rpm. In the morning, the bacterial culture was diluted 1:50 into 100 ml of semi-poor medium consisting of 95% bioassay buffer (BA: for 1 L—0.5 g Tryptone [Bacto], 0.3 ml glycerol, 5.8 g NaCl, 50 ml 1 M MgSO₄, 1 ml 10×PBS buffer pH 7.4, 950 ml DDW) and 5% LB. The inducer C4-HSL was pipetted manually to a final concentration of one out of six final concentrations: 0 μM, 0.02 μM, 0.2 μM, 2 μM, 20 μM, and 200 μM. Cells were grown at 37 °C and 250 rpm to mid-log phase (OD600 of ~0.6) as measured by a spectrophotometer and taken to the FACS for sorting.

During sorting by the FACSAria II (BD Biosciences) cell sorter, each inducer level culture was sorted into eight bins of increasing mCherry levels spanning the entire fluorescence range except for 5% at the higher end (bin 1—low mCherry to bin 8—high mCherry), and constant GFP levels (for example, the 0 mM culture were sorted according to zero GFP fluorescence, the 0.02 μM culture to slightly positive GFP fluorescence, and so on). Sorting was done at a flow rate of ~20,000 cells per second. 300k cells were collected in each bin for the entire 6 × 8 bin matrix. After sorting, the binned bacteria were transferred to 10 ml LB + Kan + Amp growth culture and shaken at 37 °C and 250 rpm overnight. In the morning, cells were prepared for sequencing (see below) and glycerol stocks were made by mixing 1 ml of bacterial solution with 500 μl 80% glycerol—20% LB solution and stored in −80 °C.

### Sequencing
Cells were lysed (TritonX100 0.1% in 1×TE: 15 μl, culture: 5 μl, 99 °C for 5 min and 30 °C for 5 min) and the DNA from each bin was subjected to PCR with a different 5' primer containing a specific bin-inducer level barcode (see Supplementary Data 5). PCR products were verified in an electrophoresis gel and cleaned using PCR clean-up kit. Equal amounts of DNA (2 ng) from 16 bins were joined to one 1.5 ml microcentrifuge tube for further analysis, to a total of three tubes. This procedure was conducted three times, one for each RBP-GFP fusion.

Each one of the three samples were sequenced on an Illumina HiSeq 2500 Rapid Reagents V2 50 bp 465 single-end chip. 20% PhiX was added as a control. This resulted in ~540 million reads, about 180 million reads per RBP.

### Bacterial cassette microscopy experiments—Addgene plasmids
pCR4-24XPP7SL was a gift from Robert Singer (Addgene plasmid #31864; http://n2t.net/addgene:31864; RRID: Addgene_31864).

### Bacterial sample preparation for microscopy
E. coli BL21-DE3 cells expressing the two plasmid systems (a single copy plasmid containing the binding site cassette, and a multicopy plasmid containing the RBP-GFP) were grown overnight in 5 ml LB, at 37 °C with appropriate antibiotics (CM, Amp), and in the presence of two inducers—1.6 μl isopropyl β-D-1-thiogalactopyranoside (IPTG) (final concentration 1 5 mM), and 2.5 μl C4-HSL (final concentration 60 μM) to induce expression of T7 RNA polymerase and the RBP-FP, respectively. Overnight culture was diluted 1:100 into 3 ml solution of (BA-LB (95%–5% v-v) with appropriate antibiotics and induced with 1 μl IPTG (final concentration 1 mM) and 1.5 μl C4-HSL (final concentration 60 μM). For stationary phase tests, cells were diluted into 3 ml Dulbecco's phosphate-buffered saline (PBS) (Biological Industries, Israel) with similar quantities of induction and 10 antibiotics. Culture was shaken for 3 h in 37 °C before being applied to a gel slide (3 ml PBS×1, mixed with 0.045 g SeaPlaque low melting agarose (Lonza, Switzerland), heated for 20 s and allowed to cool for 25 min). 1.5 μl cell culture was deposited on a gel slide and allowed to settle for an additional 30 min before imaging.

### Bacteria microscopy and analysis
Gel slide was kept at 37 °C inside an Okolab microscope incubator (Okolab, Italy). Excitation was performed at 585 nm (mCherry) wavelength by a CooLED (Andover, UK) PE excitation system. 10–15 snapshots of different fields of view (FOV) containing cells were taken for each experiment. Intensity distributions were measured using a custom Matlab (Mathworks, Natick, MA) script.

### Construction of mammalian expression plasmids
We ordered three plasmids from Addgene containing PCP-3xGFP (#75385), MCP-3xBFP (#75384), and N22-3xmCherry (#75387), and used them to create the following two plasmids: MCP-3xmCherry and QCP-3xBFP (Addgene #158206). In brief, using two restriction enzymes, BamHI and MluI, we restricted the plasmids and conducted PCR with the same restriction sites added as primers on both MCP and QCP. After PCR purification, we restricted the product with the same two enzymes and ligated them to the matching plasmids. Then, we performed transformation to Top10 E. coli cells (Lucigen) and screened for positive clones. All plasmids used in the microscopy experiments were sequence-verified via Sanger sequencing.

RNA binding site cassettes were ordered from IDT as gBlocks (see Supplementary Data 4 for sequences). We restricted and ligated them to a vector downstream of a CMV promoter using the restriction enzyme EcoRI. Then, we performed transformation to Top10 E. coli cells and screened for positive clones. All plasmids used in the microscopy experiments were sequence-verified via Sanger sequencing, and are available at Addgene (see Supplementary Data 4).

**Mammalian microscopy assay.** The human bone osteosarcoma epithelial cell line (U2OS, from ATCC) was incubated and maintained in $100 \times 20$ mm cell culture dishes under standard cell culture conditions at 37 °C in humidified atmosphere containing 5% $CO_2$ and was passaged at 80–85% confluence. Cells were washed once with 1× PBS, and subsequently treated with 1 mL trypsin/EDTA (ethylenediaminetetraacetic acid, Biological Industries) followed by incubation at 37 °C for 3–5 min. DMEMcomplete, complemented with 10% FBS and final concentrations of 100 U penicillin plus 100 μg streptomycin, was added and transferred into fresh DMEMcomplete in subcultivation ratios of 1:10.

Before the experiment, U2OS cells were seeded on 60 mm glass-bottom imaging dishes. Transient transfection was performed with Polyjet (Invivogen) transfection reagent according to the manufacturer's instructions. Typical DNA for transfection was 150 ng from RBP-3xFP and 850 ng from the cassette plasmid. After inoculation for 24–48 h, the growth medium was removed and replaced with Leibovitz L15 medium with 10% FBS. During microscopy, the sample was kept at 37 °C.

Microscopy was carried out on a Nikon Ti-E eclipse epifluorescent microscope. Images were taken with a ×40 oil immersion objective and the following excitation lasers: 585 nm for mCherry, 490 nm for GFP, 400 nm for BFP. The images were recorded with the Xion EMCCD camera. The microscope was controlled with NIS Elements imaging software. Time-lapse movies of a single Z-plane were recorded with 1500 ms exposure time and time intervals between frames were 30 s.

**Computing the responsiveness score.** Note: the following analysis procedure was conducted three times, once for each CP. Read numbers were normalized by percentage of bacteria in each bin from the total library, given by the FACS during sorting. This is done to enable comparison between numbers of reads of the same variant in different bins.

$$N_{\text{reads}}(i,j,k) = R_{\text{reads}}(i,j,k) \times \%\text{cells}(j,k), \quad \begin{matrix} i = 1:100,000 \\ j = 1:6 \\ k = 1:8 \end{matrix} \quad (1)$$

where $N_{\text{reads}}(i,j,k)$ and $R_{\text{reads}}$ are the number of normalized and raw reads per variant, bin, and inducer concentration, respectively. $\%\text{cells}(j,k)$ corresponds to the percentages of the cells in each bin per inducer concentration during sorting from the entire library as supplied by the sorter.

Two cut-offs were introduced on the variant read counts: (i) only inducer levels that had above 30 reads for all eight bins were taken into account; and (ii) only variants that had more than 300 reads in total for the entire $6 \times 8$ matrix were taken into account. For each inducer concentration $j$, we have an 8-bin histogram for which we need to calculate the mCherry averaged fluorescence of variant i $\mu(i,j)$ for all variants. First, for every variant we renormalize $N_{\text{reads}}$ by the total number of reads obtained for that inducer level (each column in the read matrix and color bar, Fig. S2a, top).

$$\tilde{N}_{\text{reads}}(i,j,k) = \frac{N_{\text{reads}}(i,j,k)}{\sum_{k=1}^{8} N_{\text{reads}}(i,j,k)}, \quad \begin{matrix} i = 1:100,000 \\ j = 1:6 \\ k = 1:8 \end{matrix} \quad (2)$$

Next, we convert the bin index ($j = 1:8$) to mCherry fluorescence ($Bin(i,j,k)$). This is done by retrieving the maximum mCherry fluorescence value that was assigned to each bin by the sorter. Then, we compute the cumulative renormalized reads by adding all the normalized reads successively from the lowest to the highest fluorescent bin as follows:

$$\tilde{N}_{\text{reads}}^{\text{cum}}(i,j,k) = \sum_{l=1}^{k} \tilde{N}_{\text{reads}}(i,j,l), \quad \begin{matrix} i = 1:100,000 \\ j = 1:6 \\ k = 1:8 \end{matrix} \quad (3)$$

Finally, to compute $\mu(i,j)$, we fit the cumulative renormalized read values to a cumulative Gaussian as follows:

$$\tilde{N}_{\text{reads}}^{\text{cum}}(i,j,k) = 0.5 + 0.5\text{erf}\left(\frac{Bin(i,j,k) - \mu(i,j)}{\sigma(i,j)\sqrt{2}}\right), \quad \begin{matrix} i = 1:100,00 \\ j = 1:6 \\ k = 1:8 \end{matrix} \quad (4)$$

where $\sigma(i,j)$ is the standard deviation for mCherry fluorescence extracted from the fitting procedure (see Fig. S2a, bottom for sample calculation). Note that only induction levels that had a goodness of fit higher than 0.5 were taken into account in the final analysis. Since each inducer concentration experiment was carried out in different conditions (e.g., duration of incubation on ice, O/N shaking, binning time) and at a different time (different days), mCherry levels assigned for each bin varied greatly as a function of experiment as well as overall fluorescence recorded. Therefore, to quantify this systematic error, we first computed a normalized mean fluorescence level ($\mu_{\text{norm}}$) per variant as follows:

$$\mu_{\text{norm}}(i,j) = \frac{\mu(i,j)}{\max\{\mu(i,j); j = 1:6\}}, \quad \begin{matrix} i = 1:100,000 \\ j = 1:6 \end{matrix} \quad (5)$$

To ascertain the scope of the problem presented by the systematic error, we plot in Fig. S2b a heat-map of $\mu_{\text{norm}}$ values consisting of 3,000 variants for PCP. Here, low fluorescence was recorded for induction levels 1, 4, and 6, while higher levels were recorded for induction levels 2, 3, and 5, respectively. These results are

consistent with the fact that the induction experiments of level 1, 4, and 6 were carried out on the same day, while those of 2, 3, and 5 on a separate day.

Next, to accommodate for these systematic discrepancies in our data, for each inducer level we extracted the $\mu_{\text{norm}}$ for all the negative control variants that were introduced into the OL (220 variants for PCP, 160 variants for MCP and QCP). We then computed the average $\mu_{\text{norm}}$ for all negative controls per inducer level to obtain $\mu_{\text{neg}}(j)$. Finally, we rescaled all $\mu_{\text{norm}}(i,j)$ values by $\mu_{\text{neg}}(j)$ to eliminate the systematic error from the average fluorescence level as follows:

$$\tilde{\mu}_{\text{norm}}(i,j) = \frac{\mu_{\text{norm}}(i,j)}{\mu_{\text{neg}}(j)}, \quad \begin{matrix} i = 1:100,000 \\ j = 1:6 \end{matrix} \quad (6)$$

Figure S2c shows that this rescaling operation successfully compensated for the systematic error. Note, that since the experiment is based on detecting a repression effect as a function of inducer, we filtered out the variants that displayed averaged mCherry levels at the three lowest concentrations below 15% of the averaged mCherry levels at the three lowest concentrations of the positive control.

To characterize binding to our variants, we compute an empirical score to quantify how similar a given variant's mCherry levels were to either the positive or negative controls. The score, termed the responsiveness score ($R_{\text{score}}$), is proportional to the binding affinity $K_d$ (see below) provided that the $R_{\text{score}}$ obtained for the various negative and positive controls are distributed in a Gaussian fashion. Quantile-quantile (QQ) plots for testing how our positive and negative controls fit to a Gaussian distribution are presented in Fig. S10.

To derive an expression for the $R_{\text{score}}$, we first compute two n-dimensional probability density functions defining the probability in an n-dimensional space to find either the CP binding or non-binding positive and negative controls, respectively. The parameters were selected according to the maximum likelihood criterion.

$$\text{pdf}(\text{pos}, n) = \frac{\exp\left(-\frac{1}{2}\left(\tilde{\mu}_{\text{norm}}(\text{pos}, n) - \text{mean}(\tilde{\mu}_{\text{norm}}(\text{pos}, n))\right)^T \Sigma^{-1} \left(\tilde{\mu}_{\text{norm}}(\text{pos}, n) - \text{mean}(\tilde{\mu}_{\text{norm}}(\text{pos}, n))\right)\right)}{\sqrt{(2\pi)^3 |\Sigma|}}$$

$$\text{pos} = \text{positive controls}$$
$$n = n_1, n_2, \ldots, n_N \quad (7)$$

$$\text{pdf}(\text{neg}, n) = \frac{\exp\left(-\frac{1}{2}\left(\tilde{\mu}_{\text{norm}}(\text{neg}, n) - \text{mean}(\tilde{\mu}_{\text{norm}}(\text{neg}, n))\right)^T \Sigma^{-1} \left(\tilde{\mu}_{\text{norm}}(\text{neg}, n) - \text{mean}(\tilde{\mu}_{\text{norm}}(\text{neg}, n))\right)\right)}{\sqrt{(2\pi)^3 |\Sigma|}},$$

$$\text{neg} = \text{negative controls}$$
$$n = n_1, n_2, \ldots, n_N \quad (8)$$

Where the set $\{n_j\}$ corresponds to $N$ independent parameters by which one can describe the fluorescence measurement of each variant, and $\sum$ is the covariance matrix. For example, one such set is the six-dimensional set corresponding to the fluorescence measurements for each inducer level.

Using these probability density functions, we can compute the probability that an n-dimensional vector $i$ belongs to each of these distributions, as follows:

$$\begin{matrix} p(i, \text{pos}) \equiv p\left(\tilde{\mu}_{\text{reg}}(i, n) | \text{pdf}(\text{pos}, n)\right) \\ p(i, \text{neg}) \equiv p\left(\tilde{\mu}_{\text{reg}}(i, n) | \text{pdf}(\text{neg}, n)\right) \end{matrix}, \quad (9)$$

which allows us to define the responsiveness score ($R_{\text{score}}$) as follows:

$$R_{\text{score}}(i) \equiv \log\left(\frac{p(i, \text{pos})}{p(i, \text{neg})}\right) \quad (10)$$

A higher $R_{\text{score}}$ indicates a more likely grouping to the CP binding positive control, while a lower score indicates a more likely grouping to the non-binding negative control.

In the analysis carried out in this paper, we chose to reduce the parameter space to a 3-dimensional space consisting of the following components: the slope ($m$) and goodness of fit ($R^2$) to a simple linear fit of the rescaled fluorescence $\tilde{\mu}_{\text{norm}}(i,j)$ to inducer concentration values. The third component is the standard deviation (std) of $\tilde{\mu}_{\text{norm}}(i,j)$ computed at the three highest concentration induction bins. We term this vector:

$$\left\{\tilde{\mu}_{\text{norm}}(i,j), \quad \begin{matrix} i = 1:100,000 \\ j = 1:6 \end{matrix}\right\} \rightarrow \left\{\tilde{\mu}_{\text{reg}}(i,n), \quad \begin{matrix} i = 1:100,000 \\ n = m, R^2, \text{std} \end{matrix}\right\} \quad (11)$$

Based on the 3-dimensional space ($R^2$, $m$, and std) we conducted a multivariant Gaussian fit for the positive and negative control populations (Fig. 2), which in turn allowed us to compute the 3-dimensional pdf(pos,$n$) and pdf(neg,$n$). Finally, we computed the $R_{\text{score}}$ for each non-control variant by averaging the score over as many barcodes which passed our filters (each variant appeared in our library 5 times). The results of this computation are presented in the heatmaps of Fig. 2 and Fig. S3, which are arranged in accordance with decreasing $R_{\text{score}}$. Up to this point, we have developed the $R_{\text{score}}$ to sort the different variants, but did not dive into what it means physically or from a binding perspective. The approach relied on mapping the behavior of the positive binding controls and non-binding negative controls in some 3-dimensional parameter space, and computing the likelihood that a given variant would belong to one or the other group. The $R_{\text{score}}$ is the log of the ratio of the two computations. In principle, $R_{\text{score}}$ can be computed from any

number of probability density functions. We could have used the original 6D space consisting of the 6 inducer concentrations, or chosen any other combination. In the computation below, we will map the 6D space to a 1D space of binding affinities that can be in principle computed from each 6-vector using a Hill function fit. In the case of such a mapping, we can replace eqn. 7 and 8 in the paper with the following terms:

$$\text{pdf}(\text{pos}, n) = \frac{1}{\sigma_{\text{pos}}\sqrt{2\pi}}\exp\left(-\frac{1}{2}\left(\frac{K_d^n - K_d^{\text{pos}}}{\sigma_{\text{pos}}}\right)^2\right), \quad \begin{array}{l}\text{pos} = \text{positive controls}\\ n = n_1, n_2, \dots, n_N\end{array}$$
$$\text{pdf}(\text{neg}, n) = \frac{1}{\sigma_{\text{neg}}\sqrt{2\pi}}\exp\left(-\frac{1}{2}\left(\frac{K_d^n - K_d^{\text{neg}}}{\sigma_{\text{neg}}}\right)^2\right), \quad \begin{array}{l}\text{neg} = \text{negative controls}\\ n = n_1, n_2, \dots, n_N\end{array} \quad (12)$$

In such a case, the probability for a given variant to have a $K_d$ similar to the positive and negative control distributions is given by:

$$p(i, \text{pos}) \equiv p\left(K_d^i | \text{pdf}(\text{pos}, n)\right)$$
$$p(i, \text{neg}) \equiv p\left(K_d^i | \text{pdf}(\text{neg}, n)\right) \quad (13)$$

We can then compute $R_{\text{score}}(i)$ similar to Eq. 10 in the following manner:

$$R_{\text{score}}(i) = \log\left[\left(\frac{\sigma_{\text{neg}}}{\sigma_{\text{pos}}}\right)\exp\left(-\frac{1}{2}\left(\frac{K_d^i - K_d^{\text{pos}}}{\sigma_{\text{pos}}}\right)^2 + \frac{1}{2}\left(\frac{K_d^i - K_d^{\text{neg}}}{\sigma_{\text{neg}}}\right)^2\right)\right] \quad (14)$$

If we assume for simplicity that $\sigma_{\text{pos}} \sim \sigma_{\text{neg}} \sim \sigma$ we get:

$$R_{\text{score}}(i) = \frac{K_d^{\text{pos}} - K_d^{\text{neg}}}{\sigma^2}K_d^i + \frac{(K_d^{\text{neg}})^2 - (K_d^{\text{pos}})^2}{\sigma^2} \quad (15)$$

which implies that the $R_{\text{score}}(i)$ for a given variant is proportional to its $K_d$.

Finally, we note that the expressions derived in equations 14 and 15 have the following general form to a reasonable first approximation:

$$R_{\text{score}}(i) = a + bK_d^i + O\left((K_d^n)^2\right) \cong a + bK_d^i \quad (16)$$

This then allows us to convert any $R_{\text{score}}$ value to binding affinity provided that we have a reasonable approximation of a and b.

Given the fact that:

$$\Delta G = -k_B T \ln K_d, \quad (17)$$

the binding energy can be estimated from $R_{\text{score}}$ values. We next used a previous study[17], which derived the $\Delta\Delta G$ for MCP with over 100k variants, 609 of them were present in our OL variants. We screened for the high affinity variants by setting thresholds of $\Delta\Delta G > -6.667$ and $R_{\text{score}} > 3.5$, which left us with 37 data points. In order to derive the $\Delta\Delta G$ for PCP and QCP using the same equation, we normalized the $R_{\text{score}}$ values by the mean calculated value for the MS2-WT strain. We then implemented a linear regression, as presented in Fig. S11, and derived a and b. Using these values, we were able to calculate $\Delta\Delta G$ for every high-affinity variant with all three RBPs. The results of this computation are given in Supplementary Data 1.

$$\Delta\Delta G(i) = \ln\frac{\frac{R.\text{score}(i)}{R.\text{score}(wt)} - a}{b}, i = 1 : 100,000 \quad (18)$$

In order to validate the Gaussian-parametric approach in our analysis, we retraced our steps and carried out a simple non-parametrized computation, called average nearest neighbor (ANN). In this case, each variant is characterized by a 6-dimensional vector representing the mean mCherry fluorescence for six inducer concentrations. For each variant, we then computed the average squared Euclidean distance in a 6-dimensional space from the positive and negative control variants, respectively, as follows:

$$S_{\text{pos}}^k = \frac{1}{N_{\text{pos}}}\sum_{i=1}^{N_{\text{pos}}}\sum_{j=1}^{6}\left(x_j^k - x_j^i\right)^2$$
$$S_{\text{neg}}^k = \frac{1}{N_{\text{neg}}}\sum_{i=1}^{N_{\text{neg}}}\sum_{j=1}^{6}\left(x_j^k - x_j^i\right)^2 \quad (19)$$

where, $x_j^k$ corresponds to the $j$th inducer concentration (varying from 1 to 6) of the $k$th variant, $x_j^i$ corresponds to the $j$th inducer concentration of the $i$th positive or negative controls variants. $N_{\text{pos}}$ and $N_{\text{neg}}$ correspond to the number of positive and

negative control variants, respectively. $S_{\text{pos}}^k$ and $S_{\text{neg}}^k$ correspond to the average squared Euclidean distance of a variant $k$ to the positive and negative control variants, respectively. We then took the logarithm of the ratio of the average squared distances (negative to positive controls—to ensure values that can correlate with parametrized $R_{\text{score}}$) to obtain a non-parametrized responsiveness score for the $k$th variant.

$$R_{\text{score}}^{\text{ANN}}(k) \equiv \log\left(\frac{S_{\text{neg}}^k}{S_{\text{pos}}^k}\right). \quad (20)$$

**Machine learning.** We developed two types of models to predict the binding preferences, represented as the responsiveness score, of the three CPs: WT-specific and whole-library. Here, we will describe in detail the models, the choice of hyper-parameters, their training on experimental data, and evaluation on a held-out test set. First, we will cover the features common to the two models. Then, we will provide details relevant to each of the two model types separately. The dataset contains $R_{\text{score}}$ of three proteins (MCP, PCP, and QCP) to ~17,000 sequences (PCP 17,177, MCP 17,213, QCP 16,041, and 12,245 in the intersection of the three). All sequences were either a variant of a known WT binding site of one of the three proteins or a non-similar sequence that was used as control (PCP 42, MCP 40, QCP 38). The edit distance of the derived sequences from their WT mostly span 4–8 mutations or indels (Fig. S1). The binding intensity score ($R_{\text{score}}$), empirically spanned the range of −281 to 47. Each sequence has a positional feature, which defines its prefix and suffix, i.e., upstream and downstream flanking sequences, respectively. The prefix is either C or GC and the corresponding suffix is one out of three options: T, CT or no suffix. The choice of suffix is done in a way that guarantees no shift in the reading frame. To provide the sequence data as input to the computational framework we used, it first needs to be transformed into numerical values. Each sequence was encoded using a traditional one-hot encoding of the sequence. Each nucleotide is converted to a four-bit vector with one bit set in the position corresponding to that nucleotide and all other positions set to zero. This way an L-long sequence is transformed into a 4xL binary matrix. L is either the WT length in the WT-specific model or 50 in the whole-library model.

We partitioned the dataset randomly into a training set (80%) and a held-out test set (20%). Then, we performed a hyper-parameters search and model training on the training set. We evaluated the trained model on the held-out test set. We used two measurements to gauge model performance: Pearson correlation and area under the receiver operating curve (AUC). Pearson correlation measures the linear agreement between two vectors, and is a common measure to evaluate intensity prediction. AUC is a common measure to evaluate the classification of positive and negative data points. We defined positive (i.e., binding) sequences as those having a binding intensity greater than 3.5, and negatives as those having intensity smaller than 3.5. This threshold was computed as the averaged $R_{\text{score}}$ of non-zero positive control variants minus one standard deviation:

Pos. control thershold =
$$\frac{\sum_1^3 \text{mean}\left(R_{\text{score}}\left(\text{pos}_{\text{control}}, i\right)\right) - \sigma\left(R_{\text{score}}\left(\text{pos}_{\text{control}}, i\right)\right)}{3}, i = \text{PCP, MCP, QCP} \quad (21)$$

We followed the hyper-parameters search and model training procedures as described in the DeepBind study[42]. First, we selected the set of hyper-parameters using 3-fold cross-validation on the training set (80% of the data). We iterated over 25 randomly selected sets of hyper-parameters from the parameter space defined for each model type (Tables 1 and 2). For each random set we performed 3-fold cross-validation on the training set. From the 25 hyper-parameters sets, we selected the set achieving maximum average Pearson correlation between predicted and measured scores of the validation fold over the 3 folds. Then, the model was trained using the selected hyper-parameters set on the training set. The final reported model evaluation is performed on the remaining 20% of the data, which serve as a held-out set not used in either the hyper-parameters search or model training. This process of parameters selection was done for each protein and for each of the models separately. For the downstream analyses performed in this study, i.e., predictions on independent datasets and for structural analysis and cassette design,

**Table 1 Hyper-parameters search space for WT-specific model.**

| Parameter space | | Final parameters (protein, prefix) | | | | | |
|---|---|---|---|---|---|---|---|
| Parameter | Initial space | MCP-C | MCP-GC | QCP-C | QCP-GC | PCP-C | PCP-GC |
| Nodes | 5–50 | 15 | 30,30 | 20 | 25 | 30,30 | 10 |
| Layers | 1–3 | 1 | 2 | 1 | 1 | 2 | 1 |
| Activation function | Identity, tanh, relu | tanh | relu | relu | relu | relu | relu |
| Epochs | 20,30..100 | 100 | 30 | 90 | 70 | 30 | 50 |

(Left) The search space of the hyper-parameters search. (Right) The hyper-parameters of the final models.

**Table 2 Hyper-parameters search for whole-library models.**

| Parameter space | | Final parameters | | |
|---|---|---|---|---|
| Parameter | Initial space | MCP | QCP | PCP |
| Nodes | 5–40 | 45 | 45 | 45 |
| Layers | 1–3 | 1 | 1 | 1 |
| Kernel length | 4–10 | 5 | 5 | 5 |
| Kernel number | 4–35 | 24 | 24 | 24 |
| Epochs | 10,20..100 | 30 | 30 | 30 |

(Left) The search space for the hyper-parameters search. (Right) The hyper-parameters of the final models.

we used a model trained and with hyper-parameters selected by 3-fold CV on the entire dataset. These processes are summarized in Fig. S12.

*WT-specific binding model.* We first tackled the challenge of developing a model based on a WT and its variants of the same length. For this aim, we used a different subset of the data for each protein. The protein-specific subset contained only the sequences that have the same length as its WT binding site (MS2—19 nt, Qβ—20 nt, PP7—25 nt). Then, we split the subset again by the prefix of the sequence (C or GC). The rationale for the second split is the low correlation in binding intensities observed between δ = C and δ = GC positions (Fig. S4). This process is summarized in Fig. 3a.

Each WT-specific model is composed of 1–2 hidden layers with 10–40 nodes and one output layer with a single node (Fig. 3a). Each protein and its sub-library have different parameters that were chosen specifically for it. This optimization process was done as described above. The details of the parameters we examined are described in Table 1. In addition to the parameters in Table 1, which are unique to each WT-specific model, there are additional parameters that are common to all of them: learning rate 0.001 (default), batch size 8, optimizer ADAM, loss function MSE (mean squared error) and dropout probability of 0.2 for each hidden layer. The output layer consisted of one node with the identity activation function.

Overall, our WT-specific models achieved good prediction performance, i.e., a Pearson correlation between 0.27 and 0.59 on a held-out test set (Fig. 3b). As explained before, the sub-library of each RBP was divided in two sub-libraries based on its prefix. We performed hyper-parameters optimization for each of the two sub-libraries and tested it on a held-out test set. For QCP a higher correlation was achieved for δ = GC, while for PCP and MCP it was achieved for δ = C.

**Whole-library binding model.** We next developed a protein-specific binding model based on the whole-library of RNA sequences and their responsiveness scores. Since the binding sites have different lengths, they need to be converted to have equal lengths for the learning process. All sequences were padded to the same length of 50 nt. The binding sites were part of an RNA transcript. Hence, we upstream-padded them with the flanking 9 or 8 nt followed by C or GC prefix (respectively) according to their position; overall 10 nt were added upstream. Downstream-padding of the sequences was done by their flanking transcriptomic context up to a full length of 50 nt.

The upstream nucleotides are:
AATTGTGAGCGCTCACAATTATGATAGATTCAATTGGATTAATTAAA GAGGAGAAAGGTACCCATG.

The downstream nucleotides are:
GTGAGCAAGGGCGAGGAGGATAACATGGCCATCATCAAGGAGTTCAT GCGCTTCAAGGTGCACATGGAGGGCTCCGTGAACGGCCACGAGTTCGA GATCGAGGGCGAGGGCGAGGGCCGCCCCTACGAGGGCACCCAGACCGC CAAGCTGAAGGTGACCAAGGGTGGCCCCCTGCCCTTCGCCTGGGACATC CTGTCCCCTCAGTTCATGTACGGCTCCAAGGCCTACGTGAAGCACC.

The padding of the binding sites does not invalidate the models. Since these flanks are constant, and the first layer of the model is a convolution layer, which extracts local sequence features, they do not have any impact on model performance.

For the whole-library binding model, we augmented the one-hot encoded sequence information by RNA secondary structure information. We used RNAfold algorithm (Vienna package[37]) to predict the structure of each sequence. The input to RNAfold is the binding site, and it outputs the predicted secondary structure in parenthesis notation, i.e., opening and closing parenthesis for base-pairs and a dot for unpaired nucleotide.

We converted this notation into an encoding of RNA structural contexts. This was done by a MATLAB script that encodes the RNA structure as a one-hot matrix with one bit set in each column for the corresponding structural context. For a binding site of length n, the n-long parenthesis annotation is transformed to a 5×n binary matrix. The structural contexts we used were lower stem (LS), bulge (B), upper stem (US), loop (L), and no-hairpin (N). The one-hot encoded structure matrix outside of the binding site was set to zero. The RNA structure matrix was concatenated to the sequence matrix (Fig. 4a). In total, for a sequence of length L, this results in a binary matrix of size $(4 + 5) \times L$.

The model is composed of one convolution layer, one hidden layer, and an output layer (Fig. 4a). The optimization of the model was done in the same manner as described above. Briefly, 25 random hyper-parameters sets were tested, and the best performing one in 3-fold cross-validation on the training set was chosen as the optimal one.

In addition to the parameters in Table 2, which are unique to each whole-library model, there are additional parameters that are common to all of them: learning rate 0.001, batch size 16, optimizer ADAM, loss function MSE (mean squared error), activation function for the convolution and hidden layers is "relu". The output layer consists of one node with the identity activation function.

The prediction performance achieved by the whole-library models is similar to the WT-specific ones, i.e., a Pearson correlation on a held-out test set greater than 0.44 for each of the three proteins (Fig. 3b). The performance as a binary classifier (motivated by the downstream application of generating non-repetitive binding site cassettes) was an AUC greater than 0.59 (Fig. S7a, empirical p-values reflecting the frequency of AUC values of random shuffles greater than the ones achieved were smaller than or equal to $10^{-3}$). In addition to achieving better Pearson correlation over the three proteins than the WT-specific models, this whole-library model has the advantage that it can be applied to a binding site of any length, and not just that of the WT. This enables the prediction of binding of all three proteins to the same sequence set.

To show case the contribution of RNA structure to our whole-library models, we compared whole-library models with and without the additional RNA structure information. We observed a slight increase in prediction performance (Fig. S7b) when the structural information was added for all three proteins. To assign statistical significance to this observation, we trained a model on 80% of the data and tested it on the remaining 20%. For each partition of the data, we performed this train and test with and without the structural context. We performed 100 repetitions of this process and evaluated the improvement using a paired Wilcoxon rank-sum test. This resulted in a significant improvement in the results when using the structural context (p-value<$10^{-6}$ for each of the three proteins).

We inspected the structural binding preferences by altering the binding site structure and predicting its binding intensity by the ML model. Three different structure alterations were made: bulge-, loop- and upper-stem-length altering mutations. To conduct this analysis in a way that is independent from sequence effects, all added nucleotides were added as a uniform vector (i.e., [0.25,0.25,0.25,0.25]).

To increase the upper-stem length, we randomly selected n positions (n = 1,2). We then inserted a base-pair with a structure context of an upper stem (i.e., A–U or C–G) to that position. Thus, we did not affect other structure elements of the binding site. Shortening of the upper stem was done by randomly deleting base-paired nucleotides. Increasing the length of the loop was done by randomly selecting n positions (n = 1,2) and inserting in that position nucleotides with the structure context of a loop. Shortening of the loop was done by randomly deleting n nucleotides from it.

Increasing bulge size was done by adding one nucleotide with the appropriate structure context. Deleting the bulge was done by simply removing the bulge nucleotide. All sequences were examined by RNAfold and showed the desired structure. The padding of these sequences was done in the same way described earlier. To test the predicted binding cassette generated according to our models' predictions, we created one million synthetic binding sites. We generated one million random sequences that are in hamming distance of 3–7 from one of the WT binding sites. Overall, we randomly selected one million out of 1.5 billion options. Because the number of possible variants rises as the length of the sequence, uniform selection of sequences will result in more variants of the long WT (PCP, 25-nt long) and less variants of the short WT (MCP, 19-nt). To overcome this bias, we divided the random selection into three parts; in each part we randomly selected 333,333 sequences from the variants of one WT. We computed the binding intensity of each of the proteins to the set of one million sequences using the whole-library models. Then, to experimentally validate model accuracy, we chose a sample out of the one million. We selected ten sequences that are single binders (i.e., bound by a single protein and not by the two others), and ten that are double binders (i.e., bound by two proteins and not by the third). As a reminder, binders are defined as having a binding score greater than 3.5, and non-binders as having a score smaller than 3.5. All are in hamming distance of at least 4 from one another and all were not included in the original experimental library.

**Reporting summary.** Further information on research design is available in the Nature Research Reporting Summary linked to this article.

## Data availability
Fasta files are available at NCBI's Sequence Read Archive (SRA). https://trace.ncbi.nlm.nih.gov/Traces/sra/?study=SRP301887. All the relevant data are available from the authors upon reasonable request. Source data are provided with this paper.

## Code availability
The software and code are publicly available: ML code and data via github.com/OrensteinLab/SynRBPbind/, https://doi.org/10.5281/zenodo.4421793. A web-tool for cassette design called CARBP is available at: https://roee-amit.technion.ac.il/our-research/software/.

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

## Acknowledgements

The authors would like to acknowledge the Technion's LS&E staff (Dr. Tal Katz-Ezov, Anastasia Diviatis) for help with sequencing and for Dr. Shay Kirzner for help with FACS-sorting.

## Author contributions

N.K. designed and carried out the OL and microscopy experiments and analysis for all constructs. E.T. wrote the code for the ML models and carried out the ML-based analysis. S.G. and O.A. assisted and guided the experiments. Z.Y. assisted with the OL design and statistical analysis of results. R.A. and Y.O. supervised the study. N.K., E.T., R.A., and Y.O. wrote the manuscript.

## Competing interests

The authors declare no competing interests.
