## [Peer Review File · Nature Communications]

Reviewers' Comments:

Reviewer #1:

Remarks to the Author:

The authors design, construct, and characterize 10000 sequence variants of protein-binding RNA hairpins that bind MS2, PP7, or Qbeta phage coat proteins by combining oligopool synthesis, library-based cloning, inducible coat protein expression, and next-generation sequencing within an inducible Sort-Seq pipeline. They converted these measurements into an outcome metric, Rscore, that was then combined with variant sequences to train a neural network model. The overall model could predict how sequence changes alter coat protein binding in terms of the Rscore metric. The authors then utilized the most non-repetitive variants of the protein-binding RNA hairpins to readily construct a 10x array and express them in mammalian cells. They used microscopy to record the movements of puncta (focal points) of coat-protein reporter fusions inside cells, confirming the functionalities of the 10x arrays. They also applied the neural network to design new protein-binding RNA hairpins that can bind either QCP/Qbeta and PCP/PP7, which successfully bound both proteins when placed in 10x arrays inside mammalian cells. Overall, the authors demonstrate the design of non-repetitive protein-binding RNA hairpins using machine learning as a guide.

However, the manuscript itself needs to be revised for both much greater clarity as well as suitability for a larger audience. Currently, the text describes the scoring, metrics, and empirical neural network modeling results in great detail. But the manuscript does not describe key results that would be expected from such a topic. There is no mention of the binding affinities or binding free energies for these protein-binding RNA hairpins (even though the experiments were set up to measure precisely these numbers). There is little mention of the role of RNA structure in determining protein affinity. The variants contain structure-altering mutations (as described), but the authors do not mention how changing the structure alters protein binding. Overall, if the analysis were improved, the work has the potential to be highly impactful.

Specific Comments

1. In the introduction, the authors attempt to propose (without much support) that kinetic and thermodynamic models are "largely qualitative" in most systems because of the absence of biological parameters. This argument is not essential to the authors' described results and it's also not correct. In their one example (predicting ribosome binding in mammalian cells), there are actually several studies quantifying and predicting how 40S ribosomes bind to the 5' methylG caps of eukaryotic mRNAs, scan across their sequences, pause in front of start codons, and initiate translation. Many of those studies do use kinetic and thermodynamic principles as their framework. Some references/examples at the end. The authors should revise their introduction to focus on the applications of protein-binding RNA hairpins across RNA synthetic biology and the challenges when attempting to build arrays of repetitive protein-binding RNA hairpins.

2. The section on calculating binding scores is very odd. Three metrics were used to calculate the authors' "Rscore", including the slope of a linear regression (repression vs. induction level), the R^2 value of the regression curve, and the standard deviation of fluorescence at the 3 highest induction bins. However, there is already a universally recognized metric for binding between macromolecular components; it's called binding affinity (an equilibrium disassociation constant). Here, the binding affinity of an RNA hairpin would be the concentration of phage coat protein that repressed reporter expression by 50%. The authors have the ability to calculate this number because they varied phage coat protein expression levels via induction and recorded the reduction in reporter expression (via Sort-Seq). They can calculate the induction level by which reporter expression levels were reduced by 50% (e.g. via interpolation or regression). Even if the authors reported their phage coat protein expression levels in terms of an equivalent inducer concentration, it would have much more meaning than this empirically defined Rscore. With their positive control RNA hairpins, the authors could even quantify the apparent binding affinities in

comparison to the wild-type RNA hairpins' binding affinities (ie, use the wild-type hairpins as a reference). Moreover, the authors may take the natural log of the binding affinity. This quantity is directly related to the RNA hairpin's binding free energy to the coat protein. Communicating such results is far easier and more impactful than reporting some arbitrary Rscore, which is meaningless to everyone else. Presumably, there is some correlation between Rscore and binding affinity. The authors should check and verify this. If true, this means their overall results should still be valid.

3. The authors mention that structure-altering mutations were introduced into several variants. However, in the "computational analysis of RNA-binding preferences" there is scant mention of how changing the hairpin structure altered protein binding. The focus is on single nucleotide changes; in single-stranded regions, structure is not presumably changed and in double-stranded regions, only small bulges are created. What happens if the structure is completely altered? How do the authors know if the variants' mutations did not completely change the RNA hairpin structure? Currently, the figures show that the RNA always forms a hairpin. That is likely not true for many variants.

4. As a demonstration, the non-repetitive 10x protein-binding RNA hairpin arrays are interesting and important. But it would be much better if these demonstrations were coupled to a scientific hypothesis or engineering objective. Perhaps the authors could investigate mRNA trafficking using the 10x arrays in a way that would be difficult to perform previously, due to the challenges of constructing/maintaining repetitive arrays? [One controversial example: Gal Haimovich et. al. "Intercellular mRNA trafficking via membrane nanotube-like extensions in mammalian cells" PNAS, DOI: 10.1073/pnas.1706365114]

5. Figure 5 has typos. Duel ==> Dual

Babendure, J. R., Babendure, J. L., Ding, J. H., & Tsien, R. Y. (2006). Control of mammalian translation by mRNA structure near caps. *Rna*, 12(5), 851-861.

Noderer, W. L., Flockhart, R. J., Bhaduri, A., de Arce, A. J. D., Zhang, J., Khavari, P. A., & Wang, C. L. (2014). Quantitative analysis of mammalian translation initiation sites by FACS-seq. *Molecular systems biology*, 10(8)

Ferreira, Joshua P., K. Wesley Overton, and Clifford L. Wang. "Tuning gene expression with synthetic upstream open reading frames." *Proceedings of the national academy of sciences* 110.28 (2013): 11284-11289.

Reviewer #2:

Remarks to the Author:

Summary

The paper proposes a high-throughput approach combining oligo libraries with machine learning, specifically neural networks, to generate a large number of new predicted binding sites for 3 phage coat proteins of bacteriophages: MCP, PCP and QCP. This approach enabled the authors to build non-repetitive RNA binding site cassettes which were tested for functionality in mammalian cells.

Recommendation

Overall, the proposed experimental approach seems promising as it could make various synthetic biology experiments more efficient by leveraging deep learning to predict new binding sites of

RBPs. However, in the current state, the paper still suffers from several serious problems on the computational side that need to be addressed, which impede the clarity of the methods as well as the reproducibility of the results. Additionally, the biological claims cannot be fully supported by the machine learning analysis given the limited predictive performance. Detailed comments and suggestions are provided below.

Major Comments

1. The abstract is rather heavy on details but seems to miss an overarching closing statement about the impact of the proposed paper. It states that: "we provide the scientific community with a novel resource for rapidly creating functional non-repetitive binding site cassettes", yet it remains unclear to the reader what the relevance or impact of this actually is.
2. The representation provided in Figure 2B should be clarified. The derivation of the multivariate Gaussian is also unclear in the method section (see comment 5 below), while the graphical representation in the form of a simple 3D scatter plot provides little additional information.
3. The description in line 568 - 571 misses any clarity - and this in a rather worrisome manner - such that it does not allow for reproduction of the results. Furthermore, I caution from using the wording 'principle component' as it can be easily misinterpreted as a principal component analysis. This section has to be clarified significantly as in my view the definition of the responsiveness score is crucial for the prediction set up.
4. In equations 8-9 (line 577 - 582) pdf's are introduced with little justification. The formulas remind of a multivariate Gaussian, yet the angle bracket notation is unusual and should be further clarified. Furthermore, when using a parametric distribution to model data (even if it is these derived fluorescence scores) two typical questions need to be addressed:
 - i) is the model justified, i.e. is the data actually Gaussian? (qq-plot etc)
 - ii) how are the pdf's parameters selected?
5. In line 625, specify how the variable length sequences were aggregated to a joint length of 50 for the whole-library model (eg. padding with zeros ..). Ideally, the authors also justify that such an aggregation does not invalidate / trivialize the task.
6. In line 614, the prediction target (responsiveness score) ranges -281 to 47, which would make the regression task the most natural prediction scenario. Yet, in line 634, an (additional) classification setup is proposed by thresholding this score.
 - i) Why are the classification labels not exhaustive in the space of possible scores? It is stated that: "We defined positive sequences as those having a binding intensity above 3.5 and negative sequences as those having a binding intensity of zero or less." What about scores between 0 and 3.5? Especially, as this endpoint tuning can have a huge impact on the final AUC, any non-obvious choice needs to be strongly justified in my view.
 - ii) In general, when converting continuous targets to distinct classes enabling classification, one has to ask the question: is there a down-stream domain-specific reason or usefulness for doing this here?
7. Parameters search (line 638 - 648): In the description provided it is unclear, and not specified, if the parameters search is performed on a validation set. Evaluating the best performing set using the same 10-fold CV would lead to overfitting. Furthermore, the subsequent fine tune and WT-specific optimisation leads to generalisation issues. Possibly, choosing the model that most commonly performs best across WT would be a better option, especially considered that in the Extended Data Figure 5 the final results are averaged.
8. Evaluation (line 673-681): The evaluation description is pretty vague "more or almost 0.6". I would recommend a more specific and detailed discussion on the performances and findings.

Additionally, as shown in Figure 5 some of the reported AUC for different proteins are very close; standard deviation should be reported in the figure to provide a better assessment and comparison of the performances. Adding to this, one could challenge the statement that an AUC of 0.6 pertains to a "good prediction performance" - given that a random prediction should get 0.5. If the authors are still convinced of the attribute "good", I would advise to justify this with non overlapping confidence bounds as compared to a random baseline.

9. Whole - library binding model: similar considerations as reported in comments (7-8) hold. In particular, looking at the result (Extended Data Figure 6) the mean AUC difference between the "with" and "without" structural information scenario is almost negligible and standard deviations are missing; such findings should be clearly discussed and reported in the manuscript.

Minor Comments

1. Nat. Commun. is a multidisciplinary journal. However, the abstract and introduction seem to be targeted to a narrower, synthetic biology audience. Specifically, the first introductory sentence (line 43 - 46) is not verbose enough, but assumes that the reader is familiar with the setup and goals of synthetic bio as well as the issues of DBT cycles. Please rephrase, and be more explicit and introductory.

2. In the abstract, the abbreviation RBP is not introduced, please define it there (as this abbreviation is overloaded, especially in biochemistry).

3. Line 52 - 53: "... which in turn require different assumptions and parameter values to model properly using a kinetic or thermodynamic modelling approach". Please rephrase, such that it is more clear 'what' to model properly.

4. Line 95: The "OL-ML approach" is not introduced before. Maybe a better phrasing is: "We propose to apply a combined OL-ML based approach".

5. Typo in line 101: computational → computationally

6. Line 111: Rather than "We previously showed" a general phrasing as "it has been shown" is more comprehensive.

7. In line 132, please introduce the abbreviation "GFP" the first time (I think here).

8. In line 668, remove redundant empty spaces (after Data Table)

9. In line 672, please be more specific what you mean with 'linear activation function'. Should it be the identity (which is what I suspect that was meant), a modified linear function (such as ReLU etc.) or literally a linear function different from the identity.

Reviewers' comments:

Reviewer #1 (Remarks to the Author):

The authors design, construct, and characterize 10000 sequence variants of protein-binding RNA hairpins that bind MS2, PP7, or Qbeta phage coat proteins by combining oligopool synthesis, library-based cloning, inducible coat protein expression, and next-generation sequencing within an inducible Sort-Seq pipeline. They converted these measurements into an outcome metric, Rscore, that was then combined with variant sequences to train a neural network model. The overall model could predict how sequence changes alter coat protein binding in terms of the Rscore metric. The authors then utilized the most non-repetitive variants of the protein-binding RNA hairpins to readily construct a 10x array and express them in mammalian cells. They used microscopy to record the movements of puncta (focal points) of coat-protein reporter fusions inside cells, confirming the functionalities of the 10x arrays. They also applied the neural network to design new protein-binding RNA hairpins that can bind either QCP/Qbeta and PCP/PP7, which successfully bound both proteins when placed in 10x arrays inside mammalian cells. Overall, the authors demonstrate the design of non-repetitive protein-binding RNA hairpins using machine learning as a guide.

However, the manuscript itself needs to be revised for both much greater clarity as well as suitability for a larger audience. Currently, the text describes the scoring, metrics, and empirical neural network modeling results in great detail. But the manuscript does not describe key results that would be expected from such a topic. There is no mention of the binding affinities or binding free energies for these protein-binding RNA hairpins (even though the experiments were set up to measure precisely these numbers). There is little mention of the role of RNA structure in determining protein affinity. The variants contain structure-altering mutations (as described), but the authors do not mention how changing the structure alters protein binding. Overall, if the analysis were improved, the work has the potential to be highly impactful.

Author response: We thank the reviewer for this comment, which encouraged us to greatly improve our manuscript and extend our analysis and results by conducting the following modifications:

1. We added a subsection under Methods showing that the responsiveness score (R_{score}) is proportional to the binding affinity. This allowed us to convert the responsiveness score to binding free energy based on *in vitro* measurements available from public literature. We provide the approximated binding free energies to each of our variants in Table S1.
2. We added a new figure, Figure 4, focusing on the analysis of the effect of RNA structural changes on the binding affinity. We now present the following analyses for each protein:
 - Structure-conserving sequence analysis (Figure 3d)
 - Bulge size alteration from no bulge to a 2nt-long bulge (Figure 4a)

- Upper stem length analysis spanning a range of $\pm 2\text{bp}$ compared to the WT (Figure 4b)
- Loop length analysis spanning a range of $\pm 2\text{nt}$ compared to the WT (Figure 4c)

These additions to the manuscript are described throughout this file, as a response to both reviewers.

Specific Comments

1. In the introduction, the authors attempt to propose (without much support) that kinetic and thermodynamic models are “largely qualitative” in most systems because of the absence of biological parameters. This argument is not essential to the authors’ described results and it’s also not correct. In their one example (predicting ribosome binding in mammalian cells), there are actually several studies quantifying and predicting how 40S ribosomes bind to the 5’ methylG caps of eukaryotic mRNAs, scan across their sequences, pause in front of start codons, and initiate translation. Many of those studies do use kinetic and thermodynamic principles as their framework. Some references/examples at the end. The authors should revise their introduction to focus on the applications of protein-binding RNA hairpins across RNA synthetic biology and the challenges when attempting to build arrays of repetitive protein-binding RNA hairpins.

Author response: We apologize for our misrepresentation of the literature, and have included in the revised introduction an additional segment focusing on the challenges of building arrays of repetitive protein-binding RNA hairpins as suggested by the reviewer.

2. The section on calculating binding scores is very odd. Three metrics were used to calculate the authors’ “Rscore”, including the slope of a linear regression (repression vs. induction level), the R^2 value of the regression curve, and the standard deviation of fluorescence at the 3 highest induction bins. However, there is already a universally recognized metric for binding between macromolecular components; it’s called binding affinity (an equilibrium disassociation constant). Here, the binding affinity of an RNA hairpin would be the concentration of phage coat protein that repressed reporter expression by 50%. The authors have the ability to calculate this number because they varied phage coat protein expression levels via induction and recorded the reduction in reporter expression (via Sort-Seq). They can calculate the induction level by which reporter expression levels were reduced by 50% (e.g. via interpolation or regression). Even if the authors reported their phage coat protein expression levels in terms of an equivalent inducer concentration, it would have much more meaning than this empirically defined Rscore. With their positive control RNA hairpins, the authors could even quantify the apparent binding affinities in comparison to the wild-type RNA hairpins’ binding affinities (ie, use the wild-type hairpins as a reference). Moreover, the authors may take the natural log of the binding affinity. This quantity is directly related to the RNA hairpin’s binding free energy to the coat protein. Communicating such results is far easier and more impactful than reporting some arbitrary Rscore, which is meaningless to everyone

else. Presumably, there is some correlation between Rscore and binding affinity. The authors should check and verify this. If true, this means their overall results should still be valid.

Author response: We agree with the reviewer that the binding affinity or binding free energy represent a more meaningful measure by which to represent our scores. However, there were multiple considerations for our choice of analysis and representation of the results that precluded the sort of analysis suggested by the reviewer. First, unlike a lower throughput measurement (e.g. Katz et al ACS Syn Bio 2018) where the K_d can be measured directly by multiple single variant measurements at an unlimited number of inducer concentrations, with an oligo library used in SORT-Seq measurement we are ultimately limited by the number of sequence reads that can be generated. Second, the fluorescence measurements are not singular measurements, but rather it is the number of variants that are detected in a particular bin, which covers a certain fluorescence range. Thus, the degree of uncertainty is considerably larger as compared with a low-throughput single variant measurement.

We chose to limit ourselves to 6 inducer concentrations and 8 fluorescent bins per protein, which necessitated ~500 Million sequence reads in total. For each variant, we reduced these large datasets to a vector of 6 values corresponding to each inducer concentration. Unfortunately, we quickly found that 6 values were insufficient for determining the K_d directly, given the inherent noise brought on by the aforementioned limitations. In retrospect, we believe that 12 concentrations may have been enough, but this is hindsight and there was no way to determine how many concentrations would be needed *a priori* (not to mention that it would require 1B reads which is financially nearly untenable).

As a result, we opted to sort our variant data by quantifying how similar they were to the ~100 positive and negative control variants that were placed within the library. This degree of similarity is the R_{score} . This definition implies that there isn't a unique way to define this similarity. For instance, we could have used a clustering analysis (e.g., by k-means) on the higher-dimensional 6-vector, or reduce the dimensionality to 1D by estimating a K_d from a hill function fit (see revised Methods). After trial and error, we determined that the 3-vector that we used generated the most reliable results, as compared via the positive and negative controls.

That being said, we recognize the community's need to bring this back full circle to a more intuitive measurement. As a result, in the revised version of the manuscript we added two elements: 1. A mathematical derivation showing that our R_{score} is proportional to K_d . 2. An approximation for the $\Delta\Delta G$'s based on literature values, which in turn allowed us to convert the R_{score} for each variant to an estimated $\Delta\Delta G$. These estimated $\Delta\Delta G$ s are now provided in Table S1.

3. The authors mention that structure-altering mutations were introduced into several variants. However, in the "computational analysis of RNA-binding preferences" there is

scant mention of how changing the hairpin structure altered protein binding. The focus is on single nucleotide changes; in single-stranded regions, structure is not presumably changed and in double-stranded regions, only small bulges are created. What happens if the structure is completely altered? How do the authors know if the variants' mutations did not completely change the RNA hairpin structure? Currently, the figures show that the RNA always forms a hairpin. That is likely not true for many variants.

Author response: We thank the reviewer for this comment, which has led us to significantly improve our manuscript. Indeed, in the original manuscript we presented a SNP-based analysis which ignored the underlying structural implications. We recognize, as the reviewer aptly pointed out, that this was insufficient.

In the revised manuscript, we now provide visualizations of 4 separate analysis: structure-conserving SNP or DNP analysis (structures in Figure 3d), structure-altering bulge analysis, upper-stem length analysis, and loop-length analysis (new figure: Figure 4). As a result, we now provide both the sequence and structural dependence of the binding specificity.

4. As a demonstration, the non-repetitive 10x protein-binding RNA hairpin arrays are interesting and important. But it would be much better if these demonstrations were coupled to a scientific hypothesis or engineering objective. Perhaps the authors could investigate mRNA trafficking using the 10x arrays in a way that would be difficult to perform previously, due to the challenges of constructing/maintaining repetitive arrays? [One controversial example: Gal Haimovich et. al. "Intercellular mRNA trafficking via membrane nanotube-like extensions in mammalian cells" PNAS, DOI: 10.1073/pnas.1706365114]

Author response: We agree with the reviewer that it would be very interesting to demonstrate the functionality of our 10x arrays by attempting to investigate a scientific problem. However, doing so would involve setting up a new project, which includes transforming our cassettes to a new system, optimizing growth and experimental conditions, conducting real-live tryouts for our newly designed cassettes, setting up a new analysis pipeline, and more. Such an effort is clearly outside of the scope of the present work, and would constitute an interesting study on its own.

5. Figure 5 has typos. Duel ==> Dual

Author response: The typo was fixed.

Babendure, J. R., Babendure, J. L., Ding, J. H., & Tsien, R. Y. (2006). Control of mammalian translation by mRNA structure near caps. *Rna*, 12(5), 851-861.

Noderer, W. L., Flockhart, R. J., Bhaduri, A., de Arce, A. J. D., Zhang, J., Khavari, P. A., & Wang, C. L. (2014). Quantitative analysis of mammalian translation initiation sites by FACS-seq. *Molecular systems biology*, 10(8)

Ferreira, Joshua P., K. Wesley Overton, and Clifford L. Wang. "Tuning gene expression with synthetic upstream open reading frames." *Proceedings of the national academy of sciences* 110.28 (2013): 11284-11289.

Reviewer #2 (Remarks to the Author):

Summary

The paper proposes a high-throughput approach combining oligo libraries with machine learning, specifically neural networks, to generate a large number of new predicted binding sites for 3 phage coat proteins of bacteriophages: MCP, PCP and QCP. This approach enabled the authors to build non-repetitive RNA binding site cassettes which were tested for functionality in mammalian cells.

Recommendation

Overall, the proposed experimental approach seems promising as it could make various synthetic biology experiments more efficient by leveraging deep learning to predict new binding sites of RBPs. However, in the current state, the paper still suffers from several serious problems on the computational side that need to be addressed, which impede the clarity of the methods as well as the reproducibility of the results.

Author response: We thank the reviewer for the comments that helped us improve the clarity of our manuscript. In the revision we have both clarified our computational pipeline and provide additional analysis using more common features, such as binding free energy, as described above in the response to reviewer #1.

Additionally, the biological claims cannot be fully supported by the machine learning analysis given the limited predictive performance.

Author response: To strengthen the validity of our computational results, we conducted statistical significance tests to gauge the predictive performance. In addition, we modified the hyper-parameter search to make it more general. Moreover, we thoroughly analyzed RNA structural binding preferences of the three tested proteins. Given that the predictions led us to infer sequence and structural RNA-binding preferences, both known and novel, and they were successfully applied in a microscopy experiment, we believe that the machine learning analysis can now support the biological claims.

Detailed comments and suggestions are provided below.

Major Comments

1. The abstract is rather heavy on details but seems to miss an overarching closing statement about the impact of the proposed paper. It states that: “we provide the scientific community with a novel resource for rapidly creating functional non-repetitive binding site cassettes”, yet it remains unclear to the reader what the relevance or impact of this actually is.

Author response: We thank the reviewer for this comment and have altered the abstract accordingly.

2. The representation provided in Figure 2B should be clarified. The derivation of the multivariate Gaussian is also unclear in the method section (see comment 5 below), while the graphical representation in the form of a simple 3D scatter plot provides little additional information.

Author response: We have clarified the analyses both in the Results and Methods sections. As for the 3D scatter, we believe it conveys the orthogonality of the three proteins adequately.

3. The description in line 568 - 571 misses any clarity - and this in a rather worrisome manner - such that it does not allow for reproduction of the results. Furthermore, I caution from using the wording ‘principle component’ as it can be easily misinterpreted as a principal component analysis. This section has to be clarified significantly as in my view the definition of the responsiveness score is crucial for the prediction set up.

Author response: We thank the reviewer for pointing out the lack of clarity in this important section and as a result we altered this entire part. We significantly expanded this section, and properly motivated the probability density analysis. Finally, we clarified the definition of R_{score} in the Methods section, where we demonstrate that the R_{score} can be computed in a variety of different ways and is not dependent on a particular set of “parameters”. See response below for an additional explanation.

4. In equations 8-9 (line 577 - 582) pdf’s are introduced with little justification. The formulas remind of a multivariate Gaussian, yet the angle bracket notation is unusual and should be further clarified. Furthermore, when using a parametric distribution to model data (even if it is these derived fluorescence scores) two typical questions need to be addressed:

i) is the model justified, i.e. is the data actually Gaussian? (qq-plot etc)

Author response: We allocated ~100 variants for both the positive and negative controls. For each population we computed the three responsiveness parameters, and fitted the data with the Gaussian. The results of the fit are provided below, showing that the positive and negative control data are approximated well by a Gaussian distribution.

ii) how are the pdf's parameters selected?

Author response: The R_{score} could be defined in other ways that would measure the same similarity. It quantifies the ratio of the probability that a particular set of parameters belongs to either the positive or negative control probability density functions. Therefore, the R_{score} can be computed for any choice of parameters. For instance, we could have computed a 6D pdf for the positive and negative controls, and from that estimated an R_{score} for each variant. Alternatively, we could have computed an R_{score} based on 1D pdf that consist of K_d 's.

As mentioned above in response to Reviewer #1, during the analysis process we found that the reduction to 1D pdf was insufficient and introduced noise, while a 6D analysis was visually unappealing for presentation purposes. Since we found no significant increase in noise in reduction to 3D from the original 6D dataset, we opted to choose these set of parameters for the analysis.

Minor comment: angle bracket in the formulas was changed.

5. In line 625, specify how the variable length sequences were aggregated to a joint length of 50 for the whole-library model (eg. padding with zeros ..). Ideally, the authors also justify that such an aggregation does not invalidate / trivialize the task.

Author response: The aggregation was done by adding 10 nucleotides upstream of the sequence, we added the adjacent 8 or 9 nucleotides upstream followed by GC or C, respectively, depending on the context of the variant. Additional nucleotides flanking the binding site downstream were added in a way that a 50nt-long sequence is formed. The addition of such flanks does not disturb model validity since these parts are constant, and thus cannot affect the learning or prediction process. We clarified this point in the subsection describing the whole-library model under the Methods section.

6. In line 614, the prediction target (responsiveness score) ranges -281 to 47, which would make the regression task the most natural prediction scenario. Yet, in line 634, an (additional) classification setup is proposed by thresholding this score.

i) Why are the classification labels not exhaustive in the space of possible scores? It is stated that: "We defined positive sequences as those having a binding intensity above 3.5

and negative sequences as those having a binding intensity of zero or less.” What about scores between 0 and 3.5? Especially, as this endpoint tuning can have a huge impact on the final AUC, any non-obvious choice needs to be strongly justified in my view.

Author response: We clarify that all models were trained on a regression problem of predicting the binding score. For evaluation purposes only, we report the area under the ROC curve of the list of binding sites ranked by the model’s predictions, where positive labels are assigned to sites with a score greater than 3.5, and negative labels to sites with a score smaller than 0. This is in addition to reporting the Pearson correlation of predicted and measured binding score over a test set (in 10-fold cross-validation).

The 3.5 positive control threshold was selected based on positive control variants. We calculated the averaged R_{score} of above-zero (see below) positive controls and subtracted one standard deviation. Taking the average over the three proteins produced a value of 3.13, which we rounded to 3.5, for a stricter threshold. Since the negative controls exhibited high variation, we chose zero as the threshold for negative variants for the above calculation. This was based on R_{score} calculation, wherein R_{score} values below zero represent that a variant is more likely to be in the negative control distribution than the positive control distribution. This justification was added under the Methods section.

Following the reviewer’s important concern and regardless of the reasoning behind the 3.5 threshold, we tested how our particular choice of high affinity threshold influenced the achieved AUC values. Therefore, we recalculated the AUC scores for two additional values: 2 and 5. As can be seen in the plot below, AUC values only slightly change with the threshold.

Figure 1. Comparison of AUC values achieved under different high-affinity thresholds.

ii) In general, when converting continuous targets to distinct classes enabling classification, one has to ask the question: is there a down-stream domain-specific reason or usefulness for doing this here?

Author response: Classification was used in order to classify binding sites from “binding” to “non-binding” for cassette assembly purposes. When we designed our cassettes, we only chose binding sites that were “binding” for the required RBP and “non-binding” for the other two.

7. Parameters search (line 638 - 648): In the description provided it is unclear, and not specified, if the parameters search is performed on a validation set. Evaluating the best performing set using the same 10-fold CV would lead to overfitting. Furthermore, the subsequent fine tune and WT-specific optimisation leads to generalisation issues. Possibly, choosing the model that most commonly performs best across WT would be a better option, especially considered that in the Extended Data Figure 5 the final results are averaged.

Author response: We thank the reviewer for this important comment. Indeed, the results that were previously reported in the manuscript gauge the ‘goodness of fit’ of the model, i.e. how well the model fits the data. They did not gauge the ability of the model to test new sequences, as the hyper-parameters were found on the same data.

To rectify this problem, we conducted a different analysis, as the reviewer suggested: the hyper-parameters were selected on a held-out validation set, comprising 20% of the original data. We then performed the 10-fold CV on the rest of the data and reported this value. This modification affects Figure S6 and the analysis described under the Methods section.

As for the fine-tuning procedure following the main search, as it is a systematic and well-defined procedure, it is a valid heuristic that can be applied on any given dataset. The averages reported in Figure S6 are over the 10 folds, and not over the proteins. Each protein is trained and tested independently of the others, which is a common strategy in the computational modeling of RNA-binding proteins. We clarified this point in the relevant subsections under the Methods section.

8. Evaluation (line 673-681): The evaluation description is pretty vague “more or almost 0.6”. I would recommend a more specific and detailed discussion on the performances and findings. Additionally, as shown in Figure 5 some of the reported AUC for different proteins are very close; standard deviation should be reported in the figure to provide a better assessment and comparison of the performances. Adding to this, one could challenge the statement that an AUC of 0.6 pertains to a “good prediction performance” - given that a random prediction should get 0.5. If the authors are still convinced of the attribute “good”, I would advise to justify this with non overlapping confidence bounds as compared to a random baseline.

Author response: We thank the reviewer for the comment on supporting our finding with statistical significance. First, for clarity we now write in the text the AUC values we achieved. Second, we added standard deviation error bars in the figures regarding the results of the 10-fold CV. Third, we performed a statistical significance analysis for each of the AUCs by generating an empirical null distribution. We randomly shuffled each test set of each fold 100 times, and calculated the fraction of times an AUC value resulting from the random ranking was greater than the one we achieved through our predictions. In all of the tests, the empirical p-value was below 0.05. The results have been added to Supplementary Figure S6 and the statistical analysis described under the Supplementary Information.

9. Whole - library binding model: similar considerations as reported in comments (7-8) hold. In particular, looking at the result (Extended Data Figure 6) the mean AUC difference between the “with” and “without” structural information scenario is almost negligible and standard deviations are missing; such findings should be clearly discussed and reported in the manuscript.

Author response: First, the same modifications applied in response to comments 7 and 8 to the WT-specific models and results, were applied to whole-library model and results. Second, we performed a statistical significance analysis for each of the AUCs by training and testing a model 100 times as follows: we trained a model on 80% of the data and tested it on the remaining 20%. For each partition of the data we performed the training and testing twice: with and without the structural context. We performed 100 repetitions of this process. We compared the results achieved in each partition, which resulted with a significant improvement in prediction accuracy when using the structural context (p-value < 10^{-8} , Wilcoxon rank-sum paired test). We added the p-value to the caption of Supplementary Figure S7 and described the statistical analysis under the Methods section.

Minor Comments

1. Nat. Commun. is a multidisciplinary journal. However, the abstract and introduction seem to be targeted to a narrower, synthetic biology audience. Specifically, the first introductory sentence (line 43 - 46) is not verbose enough, but assumes that the reader is familiar with the setup and goals of synthetic bio as well as the issues of DBT cycles. Please rephrase, and be more explicit and introductory.

Author response: The abstract and introduction were both rephrased to better suit a multidisciplinary audience. We thank the reviewer for this comment.

2. In the abstract, the abbreviation RBP is not introduced, please define it there (as this abbreviation is overloaded, especially in biochemistry).

Author response: This error was fixed.

3. Line 52 - 53: "... which in turn require different assumptions and parameter values to model properly using a kinetic or thermodynamic modelling approach". Please rephrase, such that it is more clear 'what' to model properly.

Author response: This entire part was deleted.

4. Line 95: The "OL-ML approach" is not introduced before. Maybe a better phrasing is: "We propose to apply a combined OL-ML based approach".

Author response: This phrase was taken out of the introduction.

5. Typo in line 101: computational → computationally

Author response: This error was fixed.

6. Line 111: Rather than "We previously showed" a general phrasing as "it has been shown" is more comprehensive.

Author response: This error was fixed.

7. In line 132, please introduce the abbreviation "GFP" the first time (I think here).

Author response: This error was fixed.

8. In line 668, remove redundant empty spaces (after Data Table)

Author response: This error was fixed.

9. In line 672, please be more specific what you mean with 'linear activation function'. Should it be the identity (which is what I suspect that was meant), a modified linear function (such as ReLU etc.) or literally a linear function different from the identity.

Author response: This error was fixed. We changed 'linear activation' to 'identity' throughout the manuscript.

Reviewers' Comments:

Reviewer #1:

Remarks to the Author:

Overall, the manuscript revisions have not sufficiently supported the authors' conclusions, which are quite broad in scope. There are three major deficiencies, which are correctable in theory, but perhaps not in this review process.

Specific Comments:

1. The authors focus their introduction and challenge statement on that "biological parts are often noisy and error-prone, requiring many iterations of trial and error, termed the design-build-test (DBT) cycle, before a successful design is achieved." The authors' proposed solution is to "combine high-throughput experimental techniques with machine learning (ML) algorithms" for a "big-data" solution that can generate predictive capability to overcome the DBT bottleneck". This might be an ideal general solution, but the authors' actual results are much more narrow in scope.

The dataset of 20,000 MS2, PP7, and Qbeta binding sites, characterized using iSort-Seq measurements, was used to train a convolutional neural network to qualitatively predict binding site strength. However, the authors do not test the CNN model predictions on unseen data, for example, by forward-engineering novel MS2, PP7, and Qbeta binding sites after the model training-testing process. If the authors would like to demonstrate that CNNs can be used to make generalized predictions across the entire sequence space, then additional experimental validation on unseen sequences would be necessary. Further, compared to the cornucopia of genetic parts and their interactions within genetic systems, these binding sites are relatively short RNA regions with a small sequence space and a single, predominant biomolecular interaction. The authors should not attempt to broaden their overall conclusions, as they do in their Discussion section, to encompass all RNA genetic parts or bypassing the DBT-cycle in general.

2. Regarding the microscopy measurements, the authors need to characterize additional negative controls, such as cells that do not express 10xbinding site arrays, in order to confirm that the observed fluorescent puncta are derived from the arrays. It is possible that such puncta could form without the presence of the 10xbinding arrays.

3. While the toolbox of non-repetitive MS2, PP7, and Qbeta binding sites are potentially useful, the authors have not demonstrated an application suitable for the Nature Communications audience. The authors carry out microscopy on cells expressing 10xbinding site arrays, but they do not explain why it is necessary or advantageous to utilize these 10xbinding site arrays. Do these arrays significantly improve quantification of mRNA levels or spatial resolution of transcript locations? The authors need to demonstrate some objective that required combining 10 non-repetitive MS2, PP7, and Qbeta binding sites into an array and expressing them inside cells. Otherwise, what was the compelling need to develop a machine learning model to design/predict non-repetitive MS2, PP7, and Qbeta binding sites? For example, what were the distribution of puncta across cells? What were the dynamics of puncta movement? How are these transcript dynamics affected by stress induced conditions (e.g. chemical inhibitors)? It would be straightforward to identify a relevant question that could be better addressed using the authors' toolbox of non-repetitive MS2, PP7, and Qbeta binding sites and fluorescent RNA-binding proteins.

Minor Comments:

It is probably a typo, but the sentence "In a recent study, we demonstrated model-based functional design of non-repetitive sgRNA cassettes for targeting multiple metabolic genes in bacteria [Ref 11]" is citing an article that was not written by the authors.

Reviewer #2:

Remarks to the Author:

The authors addressed several points, thank you for that. However, the following 5 major points ("re" = referring to the enumeration of the original review points) are still problematic in my view. Of these, points 1-3 could be easily addressed, whereas with point 4 and 5 I am not so sure. From a technical perspective, the rebuttal has not sufficiently convinced me. Therefore, I would advise to either address these points fully (even though I don't know how easy it is to solve problem point 5) or to resubmit an updated version to a different journal.

Major points:

1. (re 4.) "is gaussian justified"

thank you for creating these plots, I must say I do not find them too convincing as

i) a large R^2 does not necessarily imply that the Gaussian assumption is justified (q-q plot would be more informative here - as requested already in the first review)

ii) visually, these histograms do not convincingly look gaussian (in particular, see negative controls of QCP and PCP)

2. (re 4.) "how are pdf's parameters selected"

The provided answer is missing the point, I was wondering how exactly the gaussian was fit (e.g. the values in the covariance matrix are determined etc), e.g. with maximum likelihood etc. Please point me to where this is explained - in case I missed it - otherwise this would be relevant information that should not be left out.

3. (re 6.) "why are classification labels not exhaustive in the space of possible scores"

Unfortunately, varying the cut-off of 3.5 to 2 or 5 does not answer my question what happens with scores between 0 and 3.5.

I am not an expert on responsiveness scores and this specific domain application, but from an ML perspective the most natural approach would be to perform and report the regression task. In case there is a valid reason to convert this task into a binary (or even multiclass) classification task - as driven by domain-specific knowledge about biologically relevant cut-offs - it is necessary that the entire score space is accounted for in the classification labeling - otherwise, a bias is introduced to the AUC (as data with uncertain labels would be discarded). What would speak against simply taking one cut-off distinguishing between positive and negative class (or alternatively using 3 classes with one additional class representing the 'undecided' middle ground)? In any case, the frequency of each class (including the neglected class with scores from 0 to 3.5) should be stated.

4. (re 7.) "parameter search"

Thanks for adding more detailed descriptions (starting at line 728). After reading this paragraph multiple times, it unfortunately remains quite confusing.

It seems the hyperparameter search is performed using a first part of 20% of data (which again is split 10 times randomly into 80% train and 20% test). This way the best of 10 random parameter choices are determined. Then taking the remaining 80% of all data, the chosen hyperparameters are further fine-tuned in a 10-fold CV.

However, the finally reported AUC is also computed on the remaining 80% - not an with-held data split.

If I understood this part correctly, it seems that the AUC is reported on data which was trained on (so overfitting and overly optimistic / biased estimates are probable).

In case I misunderstood this part, a clearer outline would be helpful (even a schematic would be beneficial).

5. (re 8.) "good performance"

Thank you for including the significance test which shows that the performance is significantly better than random. To which degree the AUCs in the realm of 0.6 can pertain to the break-

through character of the typical Nature paper, I truly don't know as it depends on how hard to underlying task empirically is. In the end, I think this decision should be made by the biological expert or the meta-reviewer.

Minor points:

line 167: ".. only-slightly overlapping" → presenting two distinct clusters with minor overlap

line 636: ".. space- R^2 , m, and std- we .." → .. space - R^2 , m, and std - we..

Reviewer #1

1a. The authors focus their introduction and challenge statement on that “biological parts are often noisy and error-prone, requiring many iterations of trial and error, termed the design-build-test (DBT) cycle, before a successful design is achieved.” The authors’ proposed solution is to “combine high-throughput experimental techniques with machine learning (ML) algorithms” for a “big-data” solution that can generate predictive capability to overcome the DBT bottleneck”. This might be an ideal general solution, but the authors’ actual results are much more narrow in scope.

Author response: We thank the reviewer for this comment. The introduction was rewritten in the previous revision as a result of the request made by reviewer #2 to make it more accessible to a general audience.

With regards to the statement that our results are “narrow in scope”, we would like to point out to the reviewer that, to our knowledge, our work is the third study in the past year to show that using a combined machine learning and experimental high-throughput approach, the design, build, test cycle can be overcome. In brief:

1. (Reis et al. Nature Biotechnology 2019) showed that a combination of high-throughput experiments and linear discriminant analysis (LDA) on a sample dataset facilitated reliable design of predicted functional multi-sgRNA cassettes without repeat spacer sequences.
2. (de Boer et al. Nature Biotechnology 2019) went further and showed that yeast promoters with a desired regulatory output function can be reliably designed using a machine-learning model that was trained on a massively parallel reporter assay (MPRA) dataset.

Our work is the third example which shows that a large (but finite) experimental dataset can be used to train a computational model to reliably design functional biological molecules that are composite of multiple single parts. Therefore, given the success of all three examples, the conclusion that emerges is not narrow in scope, but in fact has radical implications to the future design and the understanding of biological systems.

In the final paragraph of the discussion we now add a section to discuss the implications of all three sets of findings, and we, again, thank the reviewer for this comment.

1b. If the authors would like to demonstrate that CNNs can be used to make generalized predictions across the entire sequence space, then additional experimental validation on unseen sequences would be necessary.

Author response: In the manuscript, we devoted a great deal of space for experimental verification of the machine learning model on unseen sequences. Our goal in the verification process was to achieve a higher bar of validation and provide evidence for universality. Namely, we wished to prove that our OL-ML approach produces an algorithm that is capable of creating composite parts (i.e. made of multiple characterized binding sites), and which are also applicable to other biological systems. Hence, we targeted our verification to both past experimental *in vitro* results and newly-generated experiments in mammalian cell systems, rather than remain in the original *E. coli* chassis. Specifically, we chose the following tiered verification strategy:

1. We tested our model on millions of additional MCP binding sites that were characterized by (Buenrostro et al. Nature Biotechnology 2014) *in vitro*. The correlation between the experimental values of this independent dataset and our model is shown in Fig. 5a. This was the lowest tier of validation.
2. We created a set of composite parts made of experimentally characterized binding sites. While each binding site was “seen” in the context of the OL experiment, the composite molecules were

“unseen”. Given the fact that designing DNA-based regulatory systems made of multiple characterized parts (or sequence elements) is considered to be a difficult challenge in Synthetic Biology, doing the same for functional composite RNA molecules is considered to be especially non-trivial. Therefore, successfully designing composite RNA molecules with parts that were characterized in *E.coli*, and having them successfully tested in mammalian cells corresponds to a second tier of validation. These are provided in Fig5b-e.

3. Finally, we created a composite 10x RNA molecule from untested model-generated binding sites. Moreover, this composite included only binding sites that were predicted to bind specifically both QCP and PCP (Fig. 6c). It is important to note that dual-binding coat protein binding sites neither occur in nature, nor were part of the original OL, and thus represent an entirely novel class of coat protein sites that were predicted by the ML model. A composite molecule made of a predicted new class of binding sites is the strongest possible validation, and thus corresponds to the highest tier.

Irrespective of the success of these validation experiments, we agree with the reviewer’s point that an additional verification on unseen sequences can provide important additional support to our claims. However, we believe that generating an additional dataset of iSort-seq measurements of the three proteins would have provided little value, as the measurements would have been generated by the same experimental protocol and lab (and is simulated by the reported results over 10-fold cross-validation). Validation on independent datasets, either generated by other labs or by orthogonal experimental protocols as was done for the dual-protein binding sites, is a much stronger form of verification.

Consequently, we designed another 10x cassette with purely predicted PP7 binding sites. This cassette provides another verification, and also fills a validation void in the original manuscript, as we moved directly from composite molecules made of OL-based single CP binding sites to the predicted dual-binding cassettes. The results of this validation are presented below in Fig. R1 and have been added to the SI as Fig. S9.

To summarize, we now provide three separate validations on “unseen sequences” that are shown in Figs. 5, 6, and S9, in addition to three separate validations on OL-based binding sites (Fig. 5b-e) in new “unseen” composites. We believe that six validations (three on unseen sequences and three on OL sequences) should be sufficient to prove the validity of our OL-ML approach.

Figure R1: Additional validation using 10x cassettes with unseen PP7 model-based binding sites.

1c. Further, compared to the cornucopia of genetic parts and their interactions within genetic systems, these binding sites are relatively short RNA regions with a small sequence space and a single, predominant biomolecular interaction.

Author response: We respectfully disagree with this comment. The sequence space of a 20nt binding site is 4^{20} , more than 10^{12} different sequences. Many studies have shown in the past for both DNA- and RNA-binding proteins, that simple binding models based on small datasets work to some extent. The last decade has seen a plethora of studies published in top journals measuring DNA- and RNA-binding in high-throughput assays on precisely such short sequences (including one for MS2 that was carried out *in vitro*, see Buenrostro et al. Nature Biotechnology 2014). These data made the research community realize the complexity of the binding specificity and preferences of DNA- and RNA-binding proteins (see for example Sharon et al. Nature Biotechnology 2012, Jolma et al. Cell 2013 and Weirauch et al. Nature Biotechnology 2013, among many more studies).

Moreover, RNA-binding proteins also often have structural binding preferences as for the case of phage coat proteins studied here. When including RNA structure in addition to sequence, the space of possible binding sites becomes even larger, considering that a single RNA molecule may fold into many conformations with different probabilities. This aspect significantly increases the complexity of the binding problem. Therefore, our library was designed to not only explore the sequence space of binding of the three coat proteins, but concomitantly we systematically targeted a set of structural perturbations to the binding sites. Indeed, in Figure 4 subpanels' b and c, we display the predicted binding affinity as a result of structural changes. This addition was a result of this reviewer helpful comment in the previous revision.

Finally, we previously showed (Katz et al. Cell Systems 2019) that PP7, MS2, and Q β coat-proteins generate regulatory behavior which is inconsistent with a “single, predominant biomolecular interaction”. In brief, we observed translational repression, up-regulation, and a sharply cooperative dose-response function with sites that differed by small sequence and structural mutations. Specifically, we demonstrated that deletion of two bases in the binding site alters the structure of the entire RNA molecule, thus inverting the regulatory response of the RBP from repression to activation. Thus, in order to understand how a variety of regulatory functions can be generated by coat-proteins that only have an RNA binding domain, a hybrid sequence-structural library had to be attempted.

To summarize, our work extends the oligo-library approach from a purely-sequence analysis to a wider space including structural motifs. This extension was not a mere curiosity, but rather a necessity as a result of our previous findings. Consequently, our work represents a substantial advance in both quantifying and understanding protein-RNA interactions, which will likely impact future studies in the field.

2. Regarding the microscopy measurements, the authors need to characterize additional negative controls, such as cells that do not express 10x binding site arrays, in order to confirm that the observed fluorescent puncta are derived from the arrays. It is possible that such puncta could form without the presence of the 10x binding arrays.

Author response: Since these RNA-binding proteins have been used routinely for labelling RNA during the past two decades by many groups, it is common knowledge that they do not form puncta on their own in any cell type. For this reason, we unfortunately did not add those images, and we apologize for this. The negative control images were taken and are provided below, as well as negative controls expressing CP-binding cassettes together with non-cognate RBPs (Fig. R2). None of those show puncta as expected. Those images have also been added to the SI as Fig. S8, and we thank the reviewer for this comment.

Figure R2: Negative controls. (a) Negative controls of the RBP-FP fusions transfected together with an empty pUC19 vector. (b) Images of 10x cassettes from Fig. 5e that were expressed together with non-cognate protein. (Top) 10x-MS2 together with PCP-3xGFP and QCP-3xBFP. (Middle) 10x-PP7 with MCP-3xBFP and QCP-3xBFP. (Bottom) 10x-Q β with PCP-3xGFP and MCP-3xBFP.

3a. While the toolbox of non-repetitive MS2, PP7, and Qbeta binding sites are potentially useful, the authors have not demonstrated an application suitable for the Nature Communications audience.

Author response: We respectfully disagree with this comment. We believe that this critique overlooks the fact that Synthetic Biology is a discipline in its own right. In this case, our work claims to present a solution to the DBT cycle, an incredibly thorny issue which has impeded progress in Synthetic Biology, with respect to solving the problem of synthesis, stability and transcription of DNA molecules made of repeat sequence motifs. In addition, the inability to properly synthesize DNA with sequence repeats has impeded progress in many other important biological problems (e.g. Friedreich Ataxia), and was addressed with respect to genomic-editing CRISPR applications in a recent publication from the Salis lab (see Reis et al. Nature Biotechnology 2019). Therefore, determining a systematic method by which to overcome the “repeat-problem” is not a mere curiosity of Synthetic Biologists, but rather is of interest to the wider Life-Science community. In addition, we would like to point out the following:

1. Our work and database have already garnered interest from the wider Life-Science community. Specifically, we have sent binding sites from preliminary analysis to the following labs: Elowitz (Caltech), Adelman (Harvard), and Garcia (Berkeley). In addition, we have been in contact with multiple other labs and as a result have set up a website (see <https://carbp.herokuapp.com>) to serve the community. This website is dedicated to computing user-customized cassettes, either based on the experimentally characterized variants or using the ML model.
2. We have added a third CP (QCP) to the existing repertoire, thus providing the community with a third channel that can be imaged simultaneously with PCP and MCP.
3. As discussed in the response to comment 1b above, our work provides a major step forward in the study of RBP-RNA interactions which combines sequence and structural mutation of the binding sites. Since our approach is generalizable to other RBPs, it should therefore be of interest to the wider RNA and Gene Regulatory community.

3b. The authors carry out microscopy on cells expressing 10x binding site arrays, but they do not explain why it is necessary or advantageous to utilize these 10x binding site arrays. Do these arrays

significantly improve quantification of mRNA levels or spatial resolution of transcript locations? The authors need to demonstrate some objective that required combining 10 non-repetitive MS2, PP7, and Qbeta binding sites into an array and expressing them inside cells. Otherwise, what was the compelling need to develop a machine learning model to design/predict non-repetitive MS2, PP7, and Qbeta binding sites? For example, what were the distribution of puncta across cells? What were the dynamics of puncta movement? How are these transcript dynamics affected by stress induced conditions (e.g. chemical inhibitors)? It would be straight-forward to identify a relevant question that could be better addressed using the authors' toolbox of non-repetitive MS2, PP7, and Qbeta binding sites and fluorescent RNA-binding proteins.

Author response: We agree with the reviewer that demonstrating the utility of the technology on some biologically relevant problem is important. We did so by applying our technology to the problem of liquid-liquid phase-separation in bacteria. However, it quickly became obvious that such a study is not a simple “demonstration” of the usefulness of the technology, as the reviewer requested, but rather a study on its own right. The manuscript detailing this application is presently under consideration elsewhere and is available online as a preprint (<https://www.biorxiv.org/content/10.1101/682518v2>).

The additional manuscript precisely addresses the reviewer's questions: **For example, what were the distribution of puncta across cells? What were the dynamics of puncta movement? How are these transcript dynamics affected by stress induced conditions (e.g. chemical inhibitors)?** In addition, the results presented in the follow-up manuscript also demonstrate why additional 10x cassettes are needed, as an altogether different set of conclusions was reached once the new cassettes were added. Finally, the additional manuscript clearly shows that these are not questions that can be answered in an extra panel or figure as a small addition to a larger paper. As a result, we believe that the reviewer's request is outside of the scope of the present manuscript, which details the OL-ML work.

Reviewer #2

Major points:

Author response: We thank the reviewer for highlighting the flaws of the presentation of our modelling and associated analysis. As a result, in the present revision, we present our analysis in a more compact and logically trackable fashion. Specifically, we now present a cascade of three models of increasing complexity, as follows:

1. A basic neural network (NN) model which we called WT-specific. This the sub-libraries encoded into the original OL are used as input. We present Pearson correlation values for the six sub-libraries in Fig. 3b (left and middle bars).
2. The second level is a CNN model called “whole library”, which is computed for each protein using the entire OL set of sequences (~17,000). The result of the Pearson correlation are presented in Fig. 3b (right). Note, that with the “whole-library” model PCP scores were significantly improved, while QCP and MCP remained more or less the same. We present the structure conserving results based on this model (Fig. 3d). In addition, we also use this model level to compare results from both the parametric and non-parametric R_{score} calculations (see response for point #1 below for details).
3. The third level is a whole-library CNN that incorporates RNA secondary structural information. This model (Fig. 4) is used for the analysis of the different structural mutations on CP binding. Incorporating structural information only marginally improves the Pearson correlation (see Fig. S7).

By presenting our analysis using a cascade of models characterized by Pearson correlation, we are following the standard set by (de Boer et al. Nature Biotechnology 2019) and (Urtecho et al. Biochemistry 2018). This allows us to show that our results are not an artefact of some ad-hoc choice of analysis or modelling approaches. For responses for the specific points raised, see below.

1. (re 4.) “is gaussian justified”

thank you for creating these plots, I must say I do not find them too convincing as

i) a large R^2 does not necessarily imply that the Gaussian assumption is justified (q-q plot would be more informative here - as requested already in the first review)

ii) visually, this histograms do not convincingly look gaussian (in particular, see negative controls of QCP and PCP)

Author response: We thank the reviewer for this comment and are happy to provide a response. In this case we have opted for multiple lines of evidence to support our claim.

First, we computed a qq-plot as requested (see Fig. R3 and new Figure S10). We apologize for missing this request on the initial revision). The plot shows that all positive controls and two out of the three negative controls for MCP and PCP seem to fit nicely to a Gaussian model, with only 2-3 outliers in the 0-5% quantile range.

Second, as an additional verification, we decided to retrace our steps and carry out a simple non-parametrized analysis to our data, called Average Nearest Neighbor (ANN). In this case, each variant is characterized by a 6-dimensional vector (mean mCherry fluorescence for six inducer concentrations). For each variant, we then compute the average Euclidean distance (in a six-dimensional space) from the two sets of positive and negative control variants, each set separately. We then took the logarithm of the ratio of the average distances (negative to positive controls – to ensure values that can correlate with parametrized R_{score}) to obtain a non-parametrized responsiveness score for each variant. Using this new non-parametrized approach to calculate responsiveness scores, we computed the cross-correlation with the Gaussian-parametrized R_{score} . As Figure R4 shows, for all three proteins, we obtained an average correlation coefficient of ~ 0.5 between both sets of scores. Correlation drops as set sizes increase due to the difficulty to provide robust and accurate measurements of non-binders (i.e. background variants).

Finally, we retrained the ML model using the non-parametrized scores and recomputed the binding preferences visualized on the structures as shown in Fig 3d. In Figure R5 we see that the structures computed with the non-parametrized scores are very similar to the ones computed with the Gaussian-parametrized R_{score} . While there is some deviation, because of the noisy nature of the original experimental dataset, most trends are sustained, i.e. the favored and disfavored nucleotide at each position are the same. Interestingly, with regards to comparison with our experimental data, when using the non-parameterized ANN R_{score} as training data to the CNN, Pearson correlation values of 0.42, 0.41, and 0.33 for PCP, MCP, and QCP were obtained as compared with 0.42, 0.46, and 0.44, respectively with the Gaussian-parametrized R_{score} .

To summarize, we provide a qq-plot and an independent non-parametrized analysis of the original experimental data to show that the Gaussian assumption can be considered valid, while also not affecting the ML results significantly, as a similar structural and sequence preferences emerge independent of the type of analysis.

In conclusion, we thank the reviewer again for this comment. It has allowed us to compute two different binding preferences using independently trained ML models, effectively providing us with an “error

analysis” of sorts on our experimental findings, which is not often trivial to compute with such complex datasets. Consequently, we have added Figs. R3-R5 to the paper as Figures S10, S5, and S6 respectively.

Figure R3: QQ-plot computation for the R_{score} of positive (left) and negative (right) controls.

Figure R4: (Left panels) X-Y scatter plot of the Gaussian-parametrized R_{score} (X-axis) vs the non-parametrized R_{score} . (Right panels) Cross-correlation computations between the Gaussian-parametrized and the non-parametrized R_{score} . The correlation is computed for multiple subsets of variants. Each value on the x-axis corresponds to the last-value on any subset as ordered by the Gaussian-parametrized R_{score} . Note, the correlation falls with increasing subset size due to the increased inclusion of non-binders which are expected to be randomly positioned in both the parametrized and non-parametrized spaces.

Figure R5: Comparison of structure-conserving ML mutation analysis for the non-parametrized approach (right panels) vs the Gaussian-parametrized (left panels) approach.

2. (re 4.) “how are pdf’s parameters selected”

The provided answer is missing the point, I was wondering how exactly the gaussian was fit (e.g. the values in the covariance matrix are determined etc), e.g. with maximum likelihood etc. Please point me to where this is explained - in case I missed it - otherwise this would be relevant information that should not be left out.

Author response: We apologize for our misinterpretation of the original question. The pdf’s parameters were selected according to the maximum likelihood criterion. This clarification is now added to the Methods section (pg. 14, lines 609-610).

3. (re 6.) “why are classification labels not exhaustive in the space of possible scores”

Unfortunately, varying the cut-off of 3.5 to 2 or 5 does not answer my question what happens with scores between 0 and 3.5. I am not an expert on responsiveness scores and this specific domain application, but from an ML perspective the most natural approach would be to perform and report the regression task. In case there is a valid reason to convert this task into a binary (or even multiclass) classification task - as driven by domain-specific knowledge about biologically relevant cut-offs - it is necessary that the entire score space is accounted for in the classification labeling - otherwise, a bias is introduced to the AUC (as data with uncertain labels would be discarded). What would speak against simply taking one cut-off distinguishing between positive and negative class (or alternatively using 3 classes with one additional class representing the ‘undecided’ middle ground)?

Author response: We apologize for the incomplete response to the reviewer's comment in the previous revision. As the reviewer stated, the models are trained to solve a regression problem, i.e. predicting the responsiveness score. The WT-specific models are used for mechanism understanding only, so there is indeed no valid reason to report the AUC results they achieved. As a response, we removed all reported AUC evaluations of the WT-specific models in the text and figures.

In the whole-library models we were motivated to report its performance by the AUC criteria due to the downstream application of building cassettes of non-repetitive binders, i.e. dividing the sequence space to binders and non-binders. To address the issue raised by the reviewer, we modified the definition of binders and non-binders: variants with a score above 3.5 are considered as binders, while those with a score below 3.5 are non-binders. Using this new definition, we re-calculated the AUC results reported in the manuscript in the context of the whole-library models and altered Figure S7a accordingly. In all other evaluations of the whole-library models, both in text and in figures, we removed the AUC results.

Finally, given that model performance in the literature is reported using Pearson correlation coefficients, we now present the Pearson coefficients for all models used (Fig. 3b, S7b, and at various points in the text).

In any case, the frequency of each class (including the neglected class with scores from 0 to 3.5) should be stated.

Author response: we apologize for not specifying those numbers anywhere in the manuscript. These are now mentioned in the figure caption for Fig. 2.

4. (re 7.) “parameter search”

Thanks for adding more detailed descriptions (starting at line 728). After reading this paragraph multiple times, it unfortunately remains quite confusing.

It seems the hyperparameter search is performed using a first part of 20% of data (which again is split 10 times randomly into 80% train and 20% test). This way the best of 10 random parameter choices are determined. Then taking the remaining 80% of all data, the chosen hyperparameters are further fine-tuned in a 10-fold CV.

However, the finally reported AUC is also computed on the remaining 80% - not an with-held data split.

If I understood this part correctly, it seems that the AUC is reported on data which was trained on (so overfitting and overly optimistic / biased estimates are probable).

In case I misunderstood this part, a clearer outline would be helpful (even a schematic would be beneficial).

Author response: We apologize for the unclear description in the revised manuscript. We rephrased the paragraph in the manuscript to clarify the hyper-parameter search process (see below) and added Figure Fig. R6 (Fig. S12 in the SI) for illustration.

We used a hyper-parameter search procedure, identical to the hyper-parameter search process of GraphProt (Maticzka et al. Genome Biology 2014) to optimize model performance. Given the amount of computation required for the optimization phase, all hyper-parameters were evaluated on a set of 20% of the available data. More specifically, we divided the data into two parts, 80% as training set and 20% as a validation set. Then, we randomly selected a set of parameters from the parameter space defined for each of the models (Tables S1 and S2), trained it on the training set and tested the trained model on the validation set. This step was repeated 10 times. From the 10 random parameters sets, the best performing set was selected based on the achieved Pearson correlation between predicted and measured scores of the

validation set. The second step of the search was “fine tuning” of the chosen parameter set. In this step, we tested sets of parameters in the surrounding of the set that was selected during the first step in the same manner, i.e. training the models on the training set and evaluating them the validation set. The “fine-tuning” step is based on the results of the first random stage, and thus can be generalized to any set of parameters.

The sequences used to determine the optimal parameter values, i.e. that validation set comprising of 20% of the data, were then discarded for the cross-validated performance assessment procedure. After discarding the validation set, the final reported model evaluation is by 10-fold cross validation on the remaining training set comprising of 80% of the data. This process of parameters selection was done for each protein and for each of the models separately. This process is summarized in Fig. S12 and appears below as Figure R6.

Figure R6. Schematic of the hyper-parameter search process.

5. (re 8.) “good performance”

Thank you for including the significance test which shows that the performances is significantly better than random. To which degree the AUCs in the realm of 0.6 can pertain to the breakthrough character of the typical Nature paper, I truly don't know as it depends on how hard to underlying task empirically is. In the end, I think this decision should be made by the biological expert or the meta-reviewer.

Author response: First, we would like to thank the reviewer for another constructive comment. In the manuscript, we report model performances by both AUC and Pearson correlation. We constructed a CNN-model which incorporated RNA sequence and structural information. As Fig. S7b shows, the incorporation of the structural information in the CNN improves the Pearson correlation for all three proteins by a small amount.

These numbers can be compared with the results of two relevant recent publications (de Boer et al. Nature Biotechnology 2019) and (Urtecho et al. Biochemistry 2018). In the first study, the authors used a massively parallel reporter assay (MPRA) in yeast to characterize the regulatory effect of 30 million functional variants from a fully randomized 80bp library (N80). Here, the authors trained a thermodynamic-based billboard model of transcription, which incorporated binding considerations, known TF motifs, position weighted matrices, histone occupancy, and more. They report Pearson correlation values for different transcription factors that vary from ~0.6 to 0.7 when comparing model predictions to experimental measurements. In the second study, the authors used an MPRA in bacteria to characterize the regulatory effect of ~11K promoters with different combinations of -10 and -35 elements. The authors used a neural network architecture, and achieved a Pearson correlation coefficient starting at

0.4 and increasing to 0.8 depending on the increasingly complex set of biophysical assumptions that were used to constrain the model. While these correlation coefficients are better than what we report, there are important differences that can explain the discrepancies, as follows:

1. Transcription studies using MPRA in various cell types have been carried out extensively in the past decade. This in turn has led to improved theoretical understanding of TF-binding and resultant regulatory action, which enabled the authors to construct increasingly more constrained models. As (Urtecho et al. Biochemistry 2018) show, introducing these biophysical constraints leads to large increases in predictive power, as reflected by increased R^2 .
2. By contrast, the understanding of post-transcriptional regulation is not nearly at the same level of knowledge. While binding of proteins to RNA has been studied extensively in MPRA and other high-throughput assays, the long-range structural interactions of RNAs and their effects on protein binding and subsequent translation are poorly understood. The effects are normally referred to in the literature as tertiary interactions, and there is no quantitative model at present that describes the plethora of structural interactions that occur at this level.
3. We would like to refer the reviewer to our recent publication in Cell Systems (Katz et al. Cell Systems 2019), where precisely such structural effects were studied in detail. Specifically, we demonstrated that deletion of two bases in the coat protein binding site alters the structure of the entire RNA molecule, thus inverting the regulatory response of the coat protein from inhibition to activation. As discussed in the paper, the flipping of the regulatory function on these binding sites could not be explained by a thermodynamic protein-binding model, nor could it have been predicted by state-of-the-art tools for computing RNA structure. Thus, all such effects would, by definition, fall outside the predictive scope of machine-learning models trained on conventional thermodynamic binding algorithms.
4. As a result, we only incorporated secondary or base-pairing structural considerations, which are considered state-of-the-art at the present time, but fall short when required to compute structures for long mRNA molecules in vivo. While these structural considerations improved the performance of the model, it is impossible to tell the level of performance that would be attained if we could incorporate structural models that take into account RNA tertiary interactions.

Finally, given the fact that to our knowledge there is no literature benchmark of AUC, Pearson correlation, or other metric by which to judge a post-transcriptional-based MPRA machine-learning model, the only credible metric is therefore whether or not the model was able to correctly predict regulatory or structural outcomes. In our case, the model predicted an entirely novel class of dual-binding coat protein binding sites that neither occur in nature, nor were part of the original OL. Moreover, a composite molecule made of a predicted new class of binding sites is the strongest possible validation, and represent the ultimate test. As presented in Fig. 6c, these predictions were indeed realized.

Consequently, we believe that our results of AUC which ranges between 0.57-0.65 together with a Pearson Correlation of ~0.4-0.5 should be considered sufficient, and thus should represent the benchmark for future RNA-related models. Given the importance of this comment, we have added a short statement to the discussion highlighting these results and conclusions. Finally, we would like to thank the reviewer again for these many inciteful comments, which have helped improve our manuscript immensely.

Minor points:

line 167: “.. only-slightly overlapping” → presenting two distinct clusters with minor overlap

line 636: “.. space- R^2 , m, and std- we ..” → .. space - R^2 , m, and std - we..

Authors response: The errors were corrected.

Reviewers' Comments:

Reviewer #1:

Remarks to the Author:

The authors have not adequately responded to the reviewer comments. The authors absolutely need to demonstrate some application or answer some scientific question using their 10x RBP binding site arrays in this work. To support the authors' claims of novelty, each manuscript must stand on its own. The introduction is still overly broad with speculative claims that are not supported by the authors' results. The authors' language in discussing the state of the field as well as their own results are also very "loose"; there are several statements that are incorrect or not well supported by the data. Overall, the manuscript needs to be heavily edited to focus on the authors' actual results. The authors need to be much more careful with their wording. It would be straightforward for the authors to apply their 10x RBP binding site arrays to carry out observational analysis of some phenomenon of interest, which is a suitable demonstration for this work.

Specific Comments

1. There are two distinct versions of the manuscript abstract; one on the journal website and another in the manuscript itself. The website version is succinct and focuses on the authors' results. The manuscript version is overly long, muddled with incorrect analogies, and doesn't focus on the authors' results. This reviewer considers the abstract in the manuscript document itself to be the authoritative version and so it needs to be heavily edited and changed to be much more similar to the website version.
2. As an example of loose words, in the manuscript version of the abstract, the authors describe that "biological parts are often noisy and error-prone", which doesn't make any sense. Gene expression can be considered a noisy process, but "biological parts" are not always related to gene expression. A biological part can not be "error-prone"; that's like saying a screwdriver is error-prone. It's meaningless. In particular, do the authors' really consider their RBP binding sites to be "noisy" or "error-prone"?
3. The authors' introduction still overly hypes machine learning as a panacea solution to the design-build-test challenge in synthetic biology. First, this reviewer does not agree at all with the authors' confrontational rebuttal that a protein-RNA interaction involving a 20 nt site is an ideal demonstration for how machine learning could solve all of synthetic biology's challenges. 420 (1.1×10^{12} sequences) is not a big number compared to the sequence space of a bacterial promoter ($480 = 10^{48}$), a bacterial operon ($43000 = 10^{4982}$), the DNA needed to encode a simple metabolic pathway like glycolysis ($450000 = 10^{83048}$), or an entire bacterial genome ($44600000 = 10^{7640434}$). Second, with all the data gathered, the authors' CNN has a relatively low Pearson correlation score, which severely undercuts the authors' premise that this is the ideal approach to designing genetic systems. The authors describe a Pearson R of 0.46 ($R^2 = 0.2116$) as a "strong correlation", but it is not. Again, the authors could simply treat machine learning as a tool towards discovering and identifying the determinants of RBP binding without all the hype. The authors don't need a very good machine learning model to design their 10x RBP binding site arrays (with an application), but they do need a good machine learning model to claim that machine learning is the ideal approach. The authors should extensively edit their manuscript to increase their focus on the sequence and structural determinants of the RBP binding sites as discovered & quantified by the CNN model.
4. Another example of loose words: The authors describe that the Cello algorithm utilizes "molecular mechanisms ... governed by differing kinetics, thermodynamic parameters, and free-energy considerations". In actuality, the developers of the Cello algorithm make a point about declaring that their approach relies mainly on empirical measurements of genetic circuit gate function, which is fit to phenomenological equations (e.g. the Hill equation). The authors picked a

bad example when attempting to make their point.

5. A severe case of loose words: The authors are again using the words "We demonstrated ..." in referencing an article that isn't theirs! "In a recent study, we demonstrated model-based functional design of non-repetitive sgRNA cassettes for targeting multiple metabolic genes in bacteria [Ref 11]." This was mentioned in the previous review and it was not corrected. Very sloppy.

6. The authors' entire response in "3a" is ill-formed. The authors need to carry out some application of their 10x RBP binding site arrays, which could be observational measurements of some phenomenon, to illustrate the benefit of researchers using their approach (utility). This demonstration does not need to be so fully extensive that the authors would consider it a second study. If the authors have already collected such results, then it should be straight-forward for them to include such an illustrative example. It's notably that the authors selected Reis et. al. as an example in their point. The article by Reis et. al. carries out three completely different examples of applying their non-repetitive CRISPR arrays to generate desired phenotypes. This reviewer is only asking for one such example! You're lucky.

Reviewer #2:

Remarks to the Author:

Point 1-3 were addressed sufficiently, thanks for that. Here, the only question remains: are moderate correlation coefficients slightly above 0.4 "good enough" for this high-impact journal? I am uncertain, whether this is the case. In my view, a biological domain expert must have a final say on this, I merely remain sceptical whether this performance is really convincing.

Point 4: Parameter search.

Thanks for clarifying. It makes sense to query parameters in two steps, first a rough search, and then a fine-tuned one. However, as I already indicated in the last feedback round, it seems the final performance is not reported on held-out testing data, but on the same data that was already used for training. So Figure R6 actually confirms my previous fear that overfitting is a serious issue in the current experimental setup.

If I have misunderstood the Figure R6, i.e. the reported performance was evaluated on held-out test data and not used for training before, then Figure R6 needs to be adjusted accordingly. If my interpretation is correct, however, the main results may be invalid due to overfitting.

Point 5: Thanks for the elaborations. Given that we are dealing with moderate correlation coefficients and quite moderate AUCs (not far from random), in the end the editors have to decide whether this is sufficient for publication in Nature Communications.

Overall, currently the major obstacle is the danger of overfitting due to not evaluating on properly held-out testing data. If this is truly the case (and not just misconveyed by an unfortunate figure and description), from a machine learning perspective a rejection would be warranted.

Author statement:

We thank the editors for the opportunity to resubmit a revised manuscript of our work to Nature Communications, and further appreciate the reviewer's willingness to continue reviewing our work. In the present version, we carefully edited the Introduction and Discussion sections to eliminate unclear statements, and to sharpen the focus of our conclusions on the particular coat-protein system studied in the manuscript. In addition, we re-evaluated the machine learning algorithm with the requested train/test-split approach, and demonstrated that the results achieved barely changed, and even improved, compared to the partially nested cross-validation method that we originally undertook. Thus, we hope that these corrections will alleviate the reviewers' remaining concerns.

Reviewer #1 (Remarks to the Author):

The authors have not adequately responded to the reviewer comments. The authors absolutely need to demonstrate some application or answer some scientific question using their 10x RBP binding site arrays in this work. To support the authors' claims of novelty, each manuscript must stand on its own. The introduction is still overly broad with speculative claims that are not supported by the authors' results. The authors' language in discussing the state of the field as well as their own results are also very "loose"; there are several statements that are incorrect or not well supported by the data. Overall, the manuscript needs to be heavily edited to focus on the authors' actual results. The authors need to be much more careful with their wording. It would be straightforward for the authors to apply their 10x RBP binding site arrays to carry out observational analysis of some phenomenon of interest, which is a suitable demonstration for this work.

Author response: There are two intertwined elements in the reviewer's critique. The first is the wording and the scope of the paper, and the second is the request for an application of the technology. Our paper reports a basic-science quantitative study of phage-coat protein (CP) binding to RNA. Our study adapts an oligo library and machine learning (ML) approach, which has been used successfully to characterize *E. coli* and yeast promoters in the DNA world. Our study follows the blue-print, established by those previous works and other oligo-library studies, which have all been published in high profile journals. It aims to characterize in unprecedented detail both the structural sequence determinants of CP binding to RNA. In addition, we apply a high standard for validation by directly testing predictions of the machine learning model on composite molecules containing multiple binding sites, rather than validating individual variants in a low-throughput assay. As a result, this paper provides a blue-print for how to apply the OL-ML approach to post-transcriptional regulation through the CP example. Hence, it should be of wide-interest to the general community

Since our validation strategy utilized the most common application of the technology (i.e., visualizing transcriptional processes in single cells), we understand why our work may have seemed applicative and thus incomplete to the reviewer. We apologize for this misunderstanding, and have edited the manuscript carefully to refocus the work on the basic science element.

Specific Comments

1. There are two distinct versions of the manuscript abstract; one on the journal website and another in the manuscript itself. The website version is succinct and focuses on the authors' results. The manuscript version is overly long, muddled with incorrect analogies, and doesn't focus on the authors' results. This reviewer considers the abstract in the manuscript document itself to be the authoritative version and so it needs to be heavily edited and changed to be much more similar to the website version.

Author response: The abstract in the manuscript was converted to the website version.

2. As an example of loose words, in the manuscript version of the abstract, the authors describe that "biological parts are often noisy and error-prone", which doesn't make any sense. Gene expression can be considered a noisy process, but "biological parts" are not always related to gene expression. A

biological part can not be “error-prone”; that’s like saying a screwdriver is error-prone. It’s meaningless. In particular, do the authors’ really consider their RBP binding sites to be “noisy” or “error-prone”?

Author response: Those statements were eliminated.

3. The authors’ introduction still overly hypes machine learning as a panacea solution to the design-build-test challenge in synthetic biology. First, this reviewer does not agree at all with the authors’ confrontational rebuttal that a protein-RNA interaction involving a 20 nt site is an ideal demonstration for how machine learning could solve all of synthetic biology’s challenges. 4^{20} (1.1×10^{12} sequences) is not a big number compared to the sequence space of a bacterial promoter ($4^{80} = 10^{48}$), a bacterial operon ($4^{3000} = 10^{1824}$), the DNA needed to encode a simple metabolic pathway like glycolysis ($4^{50000} = 10^{150000}$), or an entire bacterial genome ($4^{4600000} = 10^{13800000}$). Second, with all the data gathered, the authors’ CNN has a relatively low Pearson correlation score, which severely undercuts the authors’ premise that this is the ideal approach to designing genetic systems. The authors describe a Pearson R of 0.46 ($R^2 = 0.2116$) as a “strong correlation”, but it is not. Again, the authors could simply treat machine learning as a tool towards discovering and identifying the determinants of RBP binding without all the hype. The authors don’t need a very good machine learning model to design their 10x RBP binding site arrays (with an application), but they do need a good machine learning model to claim that machine learning is the ideal approach. The authors should extensively edit their manuscript to increase their focus on the sequence and structural determinants of the RBP binding sites as discovered & quantified by the CNN model.

Author response: We have extensively edited the manuscript to reflect the reviewer’s concern. That being said, the fact that a mediocre Pearson model nevertheless successfully predicted a composite molecule made of ten dual-CP-binding sites should not be trivialized. Neither us nor anyone else at present know what Pearson correlation value is needed for a model to be sufficiently reliable for molecular design. While we certainly do not claim that this is an ideal approach (and have corrected all wording which suggests otherwise), what we are saying is that given the evidence that we provide together with already published works (Urtecho et al. Biochemistry 2018, de Boer et al. Nature Biotechnology 2019, Kotopka and Smolke, Nature Communications 2020), that this is a potential pathway forward to overcoming the DBT cycle for other synthetic biological designs. In addition, while the aforementioned studies, which characterized various promoter designs in yeast and bacteria, reached a Pearson correlation coefficient of ~0.8, they did not provide an assessment of the quality of correlation that is needed for a reliable prediction of *de novo* functional sequences. Thus, our result may reflect both an insufficient understanding of protein-RNA interactions (as compared with bacterial and yeast promoters), but also that successful design may be achieved even with a “flawed” or incomplete training data for a ML-model. Either way, our results expand our ability to design CP-related systems, and provide an important contribution to the literature in the interface between Synthetic Biology and machine learning. We have substantially edited the discussion, to summarize the concerns raised by both reviewers regarding the quality of the model, and appreciate the effort taken by the reviewer to help us reach this point.

4. Another example of loose words: The authors describe that the Cello algorithm utilizes “molecular mechanisms ... governed by differing kinetics, thermodynamic parameters, and free-energy considerations”. In actuality, the developers of the Cello algorithm make a point about declaring that their approach relies mainly on empirical measurements of genetic circuit gate function, which is fit to phenomenological equations (e.g. the Hill equation). The authors picked a bad example when attempting to make their point.

Author response: That statement was eliminated from the revised introduction.

5. A severe case of loose words: The authors are again using the words “We demonstrated ...” in referencing an article that isn’t theirs! “In a recent study, we demonstrated model-based functional

design of non-repetitive sgRNA cassettes for targeting multiple metabolic genes in bacteria [Ref 11].” This was mentioned in the previous review and it was not corrected. Very sloppy.

Author response: We thank the reviewer for identifying this egregious typo!

6. The authors’ entire response in “3a” is ill-formed. The authors need to carry out some application of their 10x RBP binding site arrays, which could be observational measurements of some phenomenon, to illustrate the benefit of researchers using their approach (utility). This demonstration does not need to be so fully extensive that the authors would consider it a second study. If the authors have already collected such results, then it should be straight-forward for them to include such an illustrative example. It’s notable that the authors selected Reis et. al. as an example in their point. The article by Reis et. al. carries out three completely different examples of applying their non-repetitive CRISPR arrays to generate desired phenotypes. This reviewer is only asking for one such example! You’re lucky.

Author response: First, we would like to point out to the reviewer that the three recent publications of OL-ML studies (Urtecho et al. Biochemistry 2018, de Boer et al. Nature Biotechnology 2019, Kotopka and Smolke, Nature Communications 2020) did not provide a biological “application”, but used the machine learning model as tool to generate a deeper mechanistic and functional understanding of the promoter systems studied. Second, in the (Reis et al. Nature Biotechnology 2019) paper, a linear discriminant analysis (LDA) approach was used as a design tool to determine which nucleotides to conserve in their spacer sequence, and not as a model to generate a deeper biological understanding. Since our work belongs to the former category, despite the use of the 10x cassettes as validation tools (see response above), we respectfully disagree that an application is needed to make our work suitable for publication in Nature Communications.

Furthermore, we believe that adding an application to the paper will take away focus not only from both the structural and sequence binding specificity findings of the CPs to RNA, but also from the important discussion that needed to be had in the literature regarding the utilization and potential reliability of machine learning models in Synthetic Biology (reflecting the important discussion that was carried out in the multiple rounds of this review). Finally, since we do already possess an application and to alleviate this disagreement, we would like to request from the reviewer to let the editor decide. If the editor or another potential reviewer agrees with the request to add an application (even after this latest revision), we will do so and combine both papers.

Reviewer #2 (Remarks to the Author):

Point 1-3 were addressed sufficiently, thanks for that. Here, the only question remains: are moderate correlation coefficients slightly above 0.4 “good enough” for this high-impact journal? I am uncertain, whether this is the case. In my view, a biological domain expert must have a final say on this, I merely remain sceptical whether this performance is really convincing.

Point 4: Parameter search.

Thanks for clarifying. It makes sense to query parameters in two steps, first a rough search, and then a fine-tuned one. However, as I already indicated in the last feedback round, it seems the final performance is not reported on held-out testing data, but on the same data that was already used for training. So Figure R6 actually confirms my previous fear that overfitting is a serious issue in the current experimental setup.

If I have misunderstood the Figure R6, i.e. the reported performance was evaluated on held-out test data and not used for training before, then Figure R6 needs to be adjusted accordingly. If my interpretation is correct, however, the main results may be invalid due to overfitting.

Point 5: Thanks for the elaborations. Given that we are dealing with moderate correlation coefficients and quite moderate AUCs (not far from random), in the end the editors have to decide whether this is sufficient for publication in Nature Communications.

Overall, currently the major obstacle is the danger of overfitting due to not evaluating on properly held-out testing data. If this is truly the case (and not just misconveyed by an unfortunate figure and description), from a machine learning perspective a rejection would be warranted.

Author response: We thank the reviewer for raising the important issue of over-fitting and apologize for not responding satisfactorily in the previous revisions. In the present revision, we report evaluations of our model using results obtained on a held-out test set (i.e. neither used for training nor for hyper-parameters search). We followed a train/test split model optimization and evaluation procedure used for DeepBind (Alipanahi et al., Nature Biotechnology 2015) (Figure R1). Namely, we randomly partitioned the data into 20% serving as a held-out test set and 80% used for hyper-parameters search and model training, which we denote as the training set. To perform hyper-parameters search, we generated 25 random sets of hyper-parameters. For each set we performed 3-fold cross-validation on the training set, i.e., we split the training set into 3 equally-sized folds and repeated the following 3 times: in iteration i , fold i served as a validation set for a model trained on the two other folds. We selected the set of hyper-parameters that achieved the maximum average Pearson correlation over the 3-fold cross-validation. Then, we trained a model using the selected set of hyper-parameters on the entire training set. Last, we evaluated the model on the test set, which was held-out through the hyper-parameters search and training processes.

Figure R1. Model optimization procedure. The data is divided into 80% training set and 20% held-out test set. The hyper-parameters search process tests 25 random hyper-parameters combinations. For each combination, a 3-fold cross-validation is performed on the training set. The combination achieving the maximum average Pearson correlation is selected. This hyper-parameter set is then used in the training of a model over the training set. Last, the model is evaluated on the held-out test set.

The new model performed slightly better than the 10-CV approach that we used in prior revisions. For all three proteins the reported Pearson correlation improved by values ranging from ~5-10% (Fig. 5b). While this is not a radical improvement in performance over the 10-CV approach (consistent with prior empirical analysis of (Vabalas et al., PLOS ONE 2019)), the new approach also resulted in improved correlation with the *in vitro* MCP data reported in (Buenrostro *et al.* Nat.Biotech. 2014). Consequently, the method advocated by the reviewer not only alleviates any residual over-fitting concerns, but also resulted in a superior model.

Figure R2. Pearson correlation results for two evaluation methods: (i) the previous 10-fold cross-validation on the training set (reported as an average with error bar for standard deviation) and (ii) the new evaluation on a held-out test set.

We now list the modifications made in the manuscript. First, the subsections of model evaluation and hyper-parameters search under the Methods section were modified according to the new procedure. Second, Supplementary Figure S12 was changed according to the new parameter search and evaluation scheme (Figure R1). Third, any text reporting average correlation over 10-fold cross-validation was modified to report evaluation on a held-out test set. Fourth, the corresponding figures reporting these results (both in the main text and supplementary) were modified accordingly (Figure 2b and Figure S7). Last, we modified all figures reporting downstream analyses using the trained model. For this model, hyper-parameters search and training were performed similarly as described before, but on 100% of the data, i.e. first 3-fold CV on the entire dataset to find hyper-parameters, and then training on the entire data with the chosen hyper-parameters set. Since this model was used for down-stream analysis, i.e. predictions on independent datasets, no test data was used in the training, while utilizing 100% of the data for improved performance (as demonstrated in Figure 6d). The modified figures include structure and sequence RNA-binding preferences analysis (Figures 3d, 4b, 4c and Figure S6), comparison to (Buenrostro *et al.* Nat Biotech. 2014) *in vitro* measurements (Figure 5a), R_{score} density maps (Figure 6a), and dataset size analysis (Figure 6d).

To conclude, we thank again the reviewer for raising this important issue, and for providing us with the opportunity to optimize the performance of our model, which once again has led to a significant improvement of the manuscript.

Reviewers' Comments:

Reviewer #1:

Remarks to the Author:

The authors have adequately responded to this reviewers' comments.

As a minor point, the authors should provide a quantitative comparison (e.g. brightness or observability) between their 10x non-repetitive RBP site arrays vs. previous measurements of prior RBP arrays (perhaps with fewer sites or more repeats). Was there a functional improvement by combining 10x non-repetitive RBP sites? A typical reader will want to know this. The authors have likely have the data to answer this question already.

Reviewer #2:

Remarks to the Author:

Review Nr. 4 (3rd revision, or submission 4)

- Given that the performance has now improved since the last round (where the mediocre performance was criticised), it would have been interesting to know which changes have caused this improvement? The idea behind pointing out mediocre results in the previous iterations was not to make the authors return better-looking numbers the next time, but rather prompting them to assess the problem and to identify how the performance can be improved.

- It seems there really was a problem with overfitting which now has been addressed, the revised splitting procedure looks ok to me. Given my recommendations of the previous iteration, realizing now that this study design issue indeed was not a misunderstanding on my side, I still stand by my previous evaluation that a rejection is warranted.

Reviewer #1 query: As a minor point, the authors should provide a quantitative comparison (e.g. brightness or observability) between their 10x non-repetitive RBP site arrays vs. previous measurements of prior RBP arrays (perhaps with fewer sites or more repeats). Was there a functional improvement by combining 10x non-repetitive RBP sites? A typical reader will want to know this. The authors have likely have the data to answer this question already.

Author response: We thank the reviewer for making this request, as we do indeed have the data. We performed a comparison in *E. coli* by installing the cassettes on a single copy BAC vector. We compared new cassettes encoding 4, 5, and 10 CP binding sites to the state-of-the-art repeat cassette containing 24 PCP binding sites (Hocine *et al.*, Nat. Meth. 2013). Our result show a marked improvement in expression with comparable puncta fluorescence levels with 5 binding sites, and superior fluorescence signal with 10 binding sites. The data depicting these results were added to Figure 5 as new panel c and d.

Reviewer #2 query: Given that the performance has now improved since the last round (where the mediocre performance was criticised), it would have been interesting to know which changes have caused this improvement?"

Author response: We made several modifications in the hyper-parameters optimization process and model evaluation in the last version compared the previous version. The hyper-parameters optimization process was modified to find a robust parameters set over the training data, and model evaluation was modified to be performed on a held-out test set (as the reviewer suggested).

We list below the modifications in the order they were made and their impact on model performance.

1. Removal of the "fine tuning" step from the hyper-parameters optimization process. Averaging over the three proteins, this did not have a large affect: only a slight decrease by 0.01 Pearson correlation.
2. Increasing the number of random hyper-parameters sets. We increased the number of random sets from 10 to 25. This resulted in improved model performance: the average Pearson correlation increased by 0.02.
3. Evaluating model performance on a held-out test set. We used 64% of the data as training set, 16% as validation set (to keep the ratio between validation and training set the same as in previous steps), and 20% as a held-out test set. As expected, all models performed worse on a held-out test set: the average Pearson correlation decreased by 0.04.
4. Hyper-parameters optimization process by 3-fold cross-validation. The 3-fold cross-validation method produced a more robust parameters set and resulted in improved performance: the average Pearson correlation increased by 0.09.

Overall, we found that fine-tuning parameters by testing 25 random sets instead of 10 and by using 3-fold cross-validation instead of a validation set had a net positive effect on model performance (0.11), while the change to testing on a held-out test set had a net negative effect (0.04) which slightly reduced the over-all improvement.